# Quorum sensing integrates environmental cues, cell density and cell history to control bacterial competence

Stefany Moreno-Gámez[1,2], Robin A. Sorg[1], Arnau Domenech[1,3], Morten Kjos [1,4], Franz J. Weissing[2], G. Sander van Doorn[2] & Jan-Willem Veening [1,3]

*Streptococcus pneumoniae* becomes competent for genetic transformation when exposed to an autoinducer peptide known as competence-stimulating peptide (CSP). This peptide was originally described as a quorum-sensing signal, enabling individual cells to regulate competence in response to population density. However, recent studies suggest that CSP may instead serve as a probe for sensing environmental cues, such as antibiotic stress or environmental diffusion. Here, we show that competence induction can be simultaneously influenced by cell density, external pH, antibiotic-induced stress, and cell history. Our experimental data is explained by a mathematical model where the environment and cell history modify the rate at which cells produce or sense CSP. Taken together, model and experiments indicate that autoinducer concentration can function as an indicator of cell density across environmental conditions, while also incorporating information on environmental factors or cell history, allowing cells to integrate cues such as antibiotic stress into their quorum-sensing response. This unifying perspective may apply to other debated quorum-sensing systems.

[1] Molecular Genetics Group, Groningen Biomolecular Sciences and Biotechnology Institute, Centre for Synthetic Biology, University of Groningen, Nijenborgh 7, 9747 AG Groningen, The Netherlands. [2] Groningen Institute for Evolutionary Life Sciences, University of Groningen, P.O. Box 11103, 9700 CC Groningen, The Netherlands. [3]Present address: Department of Fundamental Microbiology, Faculty of Biology and Medicine, University of Lausanne, Biophore Building, CH-1015 Lausanne, Switzerland. [4]Present address: Department of Chemistry, Biotechnology and Food Science, Norwegian University of Life Sciences, N-1432 Ås, Norway. Correspondence and requests for materials should be addressed to G.s.v.D. (email: g.s.van.doorn@rug.nl) J.-W.V. (email: Jan-Willem. Veening@unil.ch)

Bacteria release small diffusible molecules in the extracellular medium known as autoinducers. These molecules induce the expression of particular functions including biofilm formation, luminescence and genetic competence as well as their own production[1, 2]. The most prevalent functional interpretation of the production and response to autoinducers is known as quorum sensing (QS). According to this view, the concentration of autoinducer molecules is a proxy for cell density, allowing bacteria to regulate the expression of those phenotypes that are only beneficial when expressed by many cells[1, 2]. However, it is likely that the concentration of autoinducer molecules does not only reflect cell density, but also environmental factors, such as the diffusivity of the medium. In fact, alternative hypotheses state that bacteria release autoinducers to sense these environmental factors rather than to monitor cell density. A well-known hypothesis proposed by Redfield is that the function of autoinducers is diffusion sensing, allowing cells to avoid the secretion of costly molecules under conditions where they would quickly diffuse away[3]. Other potential roles suggested for autoinducer production are sensing local cell density together with diffusion[4], the positioning of other cells during biofilm formation[5], and temporal variations in pH[6].

We study pneumococcal competence, a system classically used as an example of QS. However, whether competence is actually controlled by QS has been recently debated. Competence is a transient physiological state that is developed by *Streptococcus pneumoniae*, as well as other bacteria. Upon entry into competence, pneumococci upregulate the expression of genes required for uptake of exogenous DNA as well as bacteriocins and various genes involved in stress response[7]. In *S. pneumoniae*, competence is regulated by an autoinducer molecule known as the competence-stimulating peptide (CSP) in a two-component regulatory system formed by the histidine kinase ComD and the response regulator ComE[8, 9] (Fig. 1). Despite the detailed understanding of the regulatory network of competence induction, little is known about why competence is controlled by an autoinducer peptide like CSP. CSP has been classically thought to be a QS signal[10], whose function could be to monitor the density of potential DNA donors[11]. However, competence can be induced in response to environmental factors like pH, oxygen, phosphate, and antibiotic stress[12–15]. Based on this evidence and the finding that competence initiates at the same time in pneumococcal cultures inoculated at different initial densities, it was suggested that CSP acts as a timing device that allows cells to mount a timed response to environmental stress independently of cell density[16]. Since then, this hypothesis has established in the field as an alternative to the QS view of competence[15, 17–19]. Recently, Prudhomme et al. renamed the timing device mechanism as a growth-time dependent mechanism and proposed that a subpopulation of competent cells that originates stochastically spreads the competent state to the rest of the population by cell–cell contact[20]. Another alternative to QS is that pneumococcal competence is an instance of diffusion sensing. This was suggested by Yang et al. based on the observation that the quorum for competence induction is not fixed but decreases with more restrictive diffusion[21].

Here, we study the regulation of pneumococcal competence by cell density and two environmental factors, antibiotic stress and pH. Using batch-culture experiments, single-cell analyses, and mathematical modeling we show that these factors simultaneously regulate competence development because they all affect the CSP concentration: cell density sets the amount of cells producing CSP, whereas the environment and cell history modify the rate at which individual cells produce or sense CSP. Since there is density regulation and we show that CSP is exported extracellularly, we advocate to keep using the term

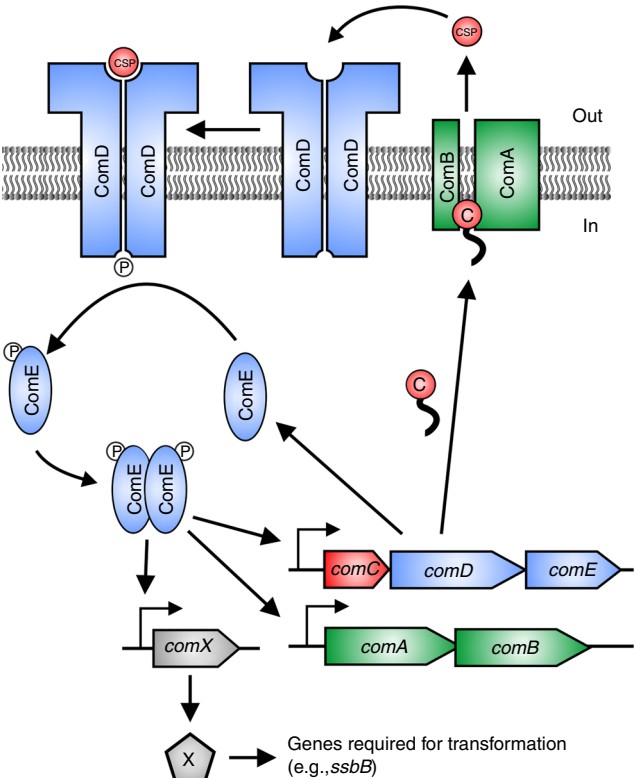

**Fig. 1** Network of competence regulation in *S. pneumoniae*. ComC (C) binds the membrane protein complex ComAB, and it is processed and exported as CSP to the extracellular space. CSP binds to the histidine kinase ComD, which is located in the membrane as a dimer. Upon CSP binding, ComD autophosphorylates and transfers the phosphate group to the response regulator ComE[9, 24]. The phosphorylated form of ComE (ComE~P) dimerizes and activates transcription of *comAB*, *comCDE*, and *comX* by binding to their promoters[8, 9]. Unphosphorylated ComE can also bind these promoters, repressing their transcription[24, 68]. Synthesis of the alternative sigma factor ComX directs transcription of genes required for genetic transformation as well as other functions[7, 25]. Two key features of this network are the presence of a positive feedback loop (since increasing CSP detection leads to increasing CSP production) and of non-linearity (since ComE-P interacts with the gene promoters as a dimer)

"quorum sensing" in the context of pneumococcal competence but with a broader meaning to acknowledge that in addition to cell density, multiple factors are integrated into this QS response.

## Results

**A mathematical model of pneumococcal competence development.** We developed a mathematical model of pneumococcal competence based on the network of protein interactions known to regulate competence development (Fig. 1) during growth in a well-mixed liquid medium. Briefly, the precursor of CSP, ComC, is cleaved and exported to the extracellular space by the membrane protein complex ComAB[9, 22]. Upon binding to CSP, ComD phosphorylates the response regulator ComE, which in its phosphorylated form upregulates transcription of the operons *comAB*, *comCDE*, and *comX*[9, 23, 24]. The latter encodes the sigma factor ComX, which controls transcription of genes required for uptake and processing of exogenous DNA[7, 25]. Our model uses ordinary differential equations (ODEs) and consists of two components. At the population level it keeps track of the population density and the extracellular concentration of CSP; at the cell level it keeps track of the intracellular concentrations

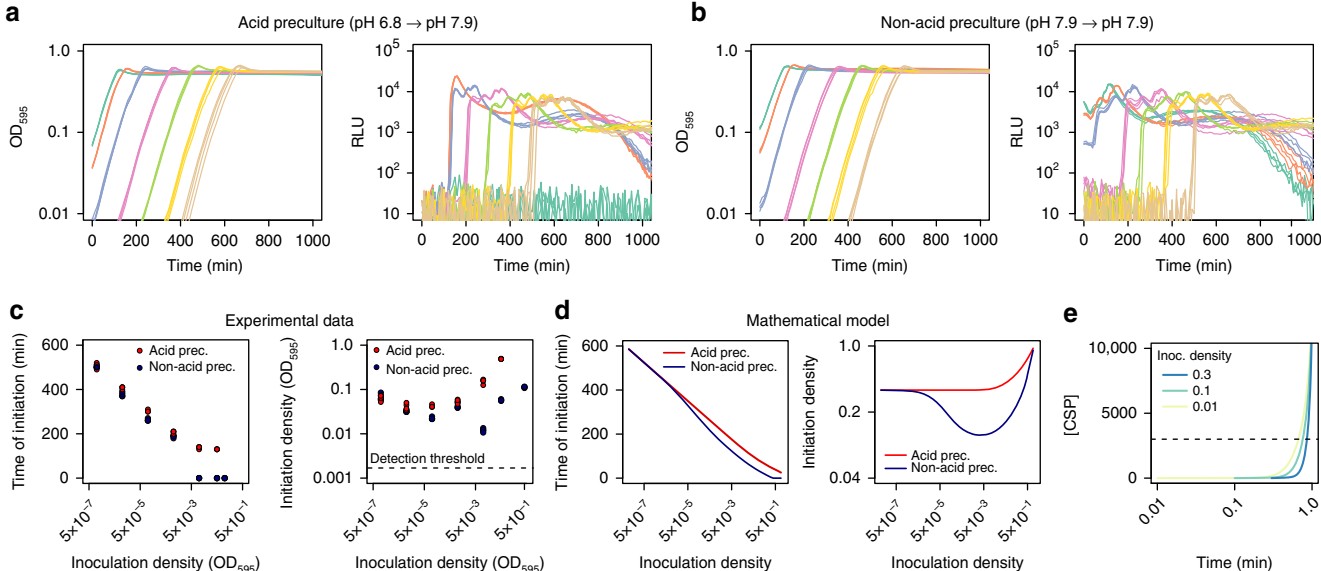

**Fig. 2** Competence is regulated by cell density. **a**, **b** Growth curves (OD$_{595 nm}$) and competence expression measured as relative luminescence units (RLU) expressed from the promoter of the late competence gene *ssbB* for populations inoculated at a range of densities and grown in C+Y medium with initial pH 7.9. In **a**, cells were precultured in acid conditions (pH 6.8), while in **b** cells were precultured in non-acid conditions (pH 7.9). Four replicates are shown for each of seven inoculation densities (OD$_{595 nm}$): 0.1 (*green*), 0.05 (*red*), 0.01 (*blue*), $10^{-3}$ (*purple*), $10^{-4}$ (*light green*), $10^{-5}$ (*yellow*), and $10^{-6}$ (*brown*). Competence does not develop in cells coming from acid preculture and inoculated at a density of 0.1. **c** Effect of inoculation density on the time until competence initiation (*left panel*) and the population density at which competence was initiated (*right panel*). Competence initiation was defined as the time where the RLU signal exceeded 200 units. Note that our luminometer has enough sensitivity to detect light from competent cells at a density of 1.56 ×$10^{-3}$ or higher even if they correspond to a subpopulation (Supplementary Fig. 2). **d** Predictions of the mathematical model concerning the effect of inoculation density on the timing of competence initiation (*left panel*) and the density at which competence initiates (*right panel*). In the model, competence initiation was defined as the time where the total concentration of ComX times the population density exceeds 2000 units. Non-acid preculture is simulated in the model by setting the initial amount of all proteins in the competence regulatory network to the value they attain when cells are competent. **e** The model predicts that populations inoculated at lower densities will reach a threshold CSP concentration (*dotted line*) at a lower density than populations inoculated at higher densities

of the proteins involved in competence regulation (Fig. 1). All the cells export CSP to the medium at a rate determined by the intracellular concentrations of ComC and ComAB. The concentration of CSP then feeds back into the intracellular concentrations of all the proteins involved in competence since their transcription rates depend on the ratio of ComE to ComE~P and thus on the rate at which ComD phosphorylates ComE. Different environmental scenarios are simulated by changing model parameters according to the known effects of such environmental factors in the competence regulatory network. Since the model is primarily concerned with competence initiation we purposely left out genes crucial for other aspects of competence development (e.g., the stabilizing factor ComW and the immunity gene *comM*)[18] and genes involved in competence shut-off such as DprA[26]. A detailed description of the model and the choice of parameter values is provided in Supplementary Note 1.

We use the model to determine the effect of environmental factors and cell history (i.e., environments experienced in the past) on the relationship between cell density and CSP concentration. Crucially, the model assumes that all cells are homogeneous and that competence is only regulated by CSP, whose production increases with cell density since cells release all CSP they produce to a common extracellular pool. We are interested in determining whether these assumptions are sufficient to explain our experimental results in well-mixed cultures or if additional mechanisms need to be incorporated (e.g., density-independent competence induction[16, 17] and cell–cell contact dependent competence transmission[20]).

**Competence develops at a critical CSP concentration**. It has been reported that competence develops at a fixed time after inoculation from acid to alkaline conditions (pH 6.8→7.9) regardless of the inoculum size[16, 20]. This observation has motivated the view that competence develops independently of cell density and rather acts as a timed response at the single-cell level to the pH shift occurring at the moment of inoculation. We extended previous studies by exploring a wider range of inoculation densities (OD$_{595 nm}$: $10^{-1}$–$10^{-7}$) (~$10^8$–$10^2$ cells per mL) and preculturing conditions. We used the encapsulated serotype 2 strain *S. pneumoniae* D39[27], and cells were washed before inoculation to remove CSP produced during the preculture. Importantly, we verified that CSP is actually present in the supernatant of competent cultures of strain D39 (Supplementary Fig. 1). To monitor competence development, the ComX-dependent promoter of the late competence gene *ssbB* was fused to the firefly *luc* gene and inserted at the non-essential *bgaA* locus. Activation and expression of *ssbB* is a good reporter for competence development since SsbB expression strongly correlates with actual transformation with externally added DNA (e.g., refs. [15, 28]).

As shown in Fig. 2a, we find that the inoculation density in strain D39 does have an effect on the time of competence development, with competence initiating later for lower inoculum sizes. For instance, for the lowest inoculation density, competence initiates more than 4 h later than for the highest inoculation densities (Fig. 2a and *left panel* of 2c). Note that our luminometer can detect light from competent cells at an OD$_{595}$ of 1.56×$10^{-3}$ or higher (Supplementary Fig. 2), and therefore we cannot exclude

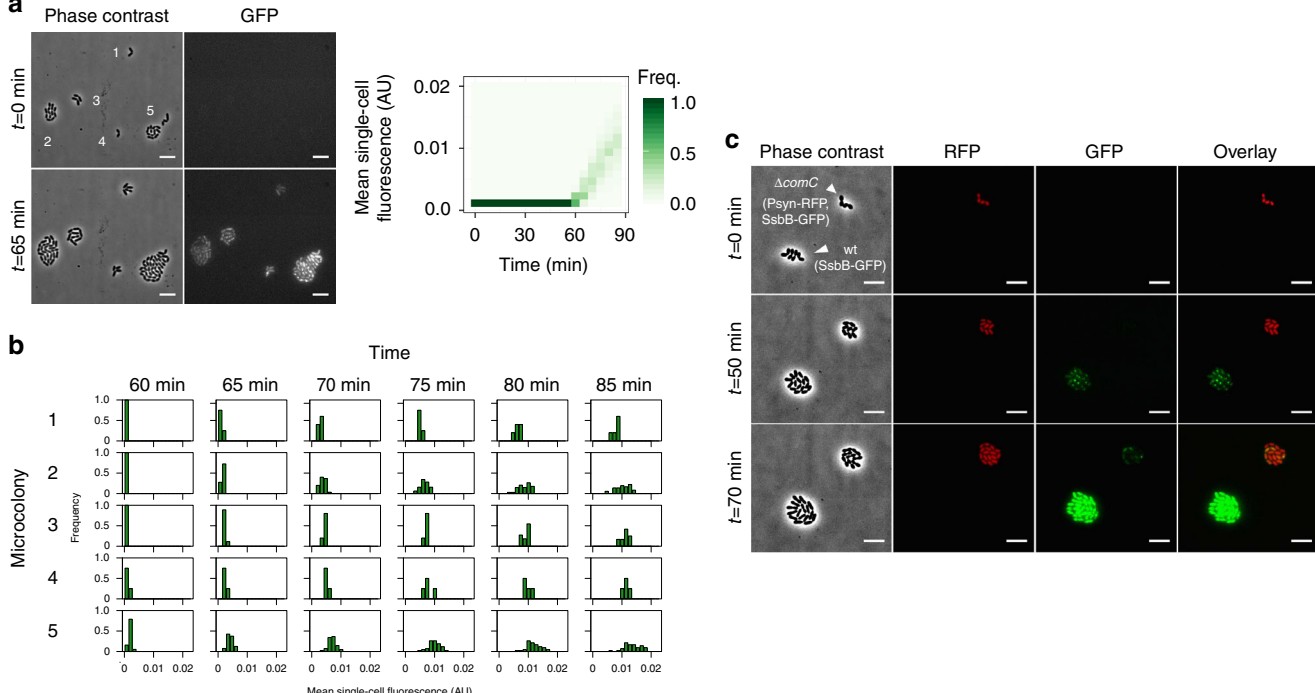

**Fig. 3** Competence can propagate by CSP diffusion without the necessity of cell-to-cell contact. **a** Phase contrast and GFP images (*left panel*) of a set of neighboring microcolonies of D39 with a fusion of the late competence gene *ssbB* to *gfp* (P*ssbB*–*ssbB*–*gfp*). The first images (*t* = 0 min) were taken right after inoculation on the microscopy slide and the second ones correspond to the moment of competence initiation (*t* = 65 min). Images were taken every 5 min. Crucially, when competence starts the microcolonies are not in direct physical contact with each other. Analysis of the fluorescence signal across the entire population (*right panel*) shows that once competence starts, after 60 min from inoculation, the distribution of fluorescence intensity moves to higher intensity values through time. We set as a threshold for counting competent cells a fluorescence signal 50% above background. After 85 min, we counted 97 out of 99 cells as competent cells. **b** Distribution of fluorescence signal within each microcolony in the window of competence initiation. The microcolony numbers correspond to the ones indicated in the phase contrast image in **a**). Competence initiates in all microcolonies within a window of 5 min. **c** Time-lapse fluorescence microscopy tracking competence development in two colonies with a fusion of the late competence gene *ssbB* to *gfp*: one formed by cells of D39 (ADP249) and one formed by cells of a *comC*-deficient D39 (ADP247). The two strains are distinguishable because ADP247 constitutively expresses a red fluorescent protein. Competence developed first in D39 and then it propagated to the *comC*⁻ mutant after 20 min without the necessity of cell–cell contact. Note that we checked that the *comC*-deficient D39 (ADP247) does not become competent by itself when grown without D39 (Supplementary Fig. 4 and Supplementary Movies 3 and 4). The *scale bar* is 5 μm in all images

the possibility that a very small subpopulation of cells initiates competence before we can detect it. Nevertheless, in all cases our estimates of the density of competence initiation are far higher than the detection threshold, indicating that competence in the majority of the population had not developed before crossing the density threshold (Fig. 2c, *right panel*).

Importantly, we observe that the population density at competence initiation is not constant but positively related to the inoculation density. Hence, the dependency of the time of competence initiation on the inoculation density is not a consequence of competence developing at a fixed critical cell density for every condition. Instead, our results are consistent with the mathematical model, which predicts that competence develops when the CSP concentration has reached a critical threshold. The model shows that competence will start faster for higher inoculation densities because the CSP concentration reaches the critical threshold for competence activation earlier if more cells are producing CSP (Fig. 2d, *left panel*). Moreover, the model shows that populations inoculated at low densities initiate competence at a lower density than populations inoculated at high densities consistent with the experimental data (*right panels* of Fig. 2c, d). This is because cells inoculated at low cell densities already had time to start transcribing competence regulatory genes and accumulate some CSP once they reached the same cell density of cultures freshly inoculated at a higher cell density

(Fig. 2e). Thus, the critical CSP threshold is reached sooner for low-density inoculated cultures. Notably, a common misconception in the field is that in a QS system the critical concentration of autoinducer should always be attained at the same fixed cell density[16, 17, 20, 21].

It is well known that the pH of the medium affects competence development, with natural competence being inhibited under acid conditions[13, 29]. Under our experimental conditions, competence only naturally develops in alkaline growth medium with a pH > 7.4. So far, we have studied competence with cells precultured in a non-permissive pH for competence development (pH 6.8). These preculture conditions were reflected in the model simulations by assuming that cells initially were in the competence-off state. We also simulated the alternative scenario that cells are already competent at inoculation. For this cell history, the model predicts that the time of competence initiation is lower, but only for high inoculation densities (Fig. 2d, *left panel*). This happens because when cells are competent initially and are inoculated at high density they can produce enough CSP to remain competent (Supplementary Fig. 3). However, when inoculation density is low, cells cannot produce enough CSP and initial competence switches off. The timing of the subsequent competence initiation is then the same as if cells were not competent when inoculated.

To verify the predicted effect of cell history on the timing of competence initiation, we controlled the competence state of cells

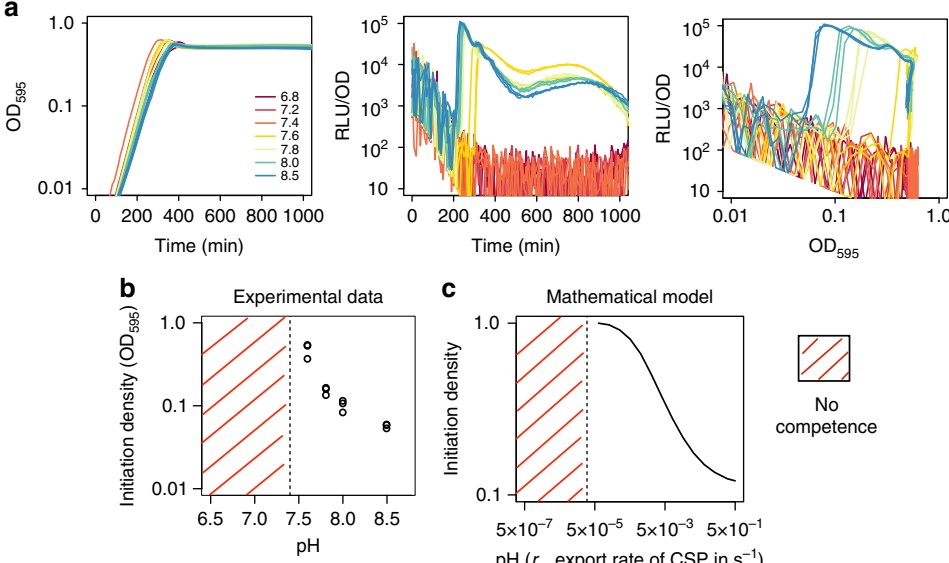

**Fig. 4** Competence is upregulated by higher pH. **a** Effect of initial medium pH on growth curves (*left panel*) and the dynamics of competence expression (*middle panel*). Competence expression was quantified as relative luminescence units (*RLU*) normalized by cell density. In the *right panel*, competence expression is plotted in relation to cell density. All populations were grown at the indicated initial pH and inoculated at a density of OD$_{595\,nm}$ 0.002. Three replicates are shown for each initial pH. **b** Effect of initial medium pH on the population density at which competence was initiated (the density at which RLU exceeded 200 units). Competence did not develop at pH 7.4 and below. Note that the indicated pH is the initial pH of the medium, which does not stay constant due to acidification by growth (Supplementary Fig. 7). Although the pH drops considerably in fully grown cultures, acidification is still minor at the density where competence develops. **c** Predictions of the model on the effect of the rate of CSP export, $r_e$, and thus the pH, on the density of competence initiation. Competence does not develop any more below a threshold rate of CSP export

at inoculation by manipulating the pH during preculture. Specifically, we compared the time of competence initiation for cells coming from a non-permissive (pH 6.8) and a permissive (pH 7.9) pH history for competence development. For inoculation densities below OD$_{595\,nm}$ of 0.01, the pH of the preculture did not have an effect on the timing of competence initiation as predicted by the model (Fig. 2b, c). On the other hand, for inoculation densities above OD$_{595\,nm}$ of 0.01, there was a time delay in competence initiation for cells with an acid history whereas cells with a non-acid history were competent when inoculated and remained competent afterwards. This suggests that when the inoculation density is high, there are enough cells to take the CSP concentration above the threshold for competence activation if they are already producing CSP—as is the case of cells coming from a non-acid history. By contrast, if cells come from a non-permissive pH for competence development, the machinery for CSP production needs to be activated. This causes a delay in competence initiation at high cell densities, which, at least for our strain, would not result from regulation by a cell-density-independent timing device[16, 17, 20].

To further study competence in conditions closer to what *S. pneumoniae* experiences in nature, we monitored competence development at the single-cell level in microcolonies growing on a semi-solid surface. To this end, *ssbB* was fused to *gfp* and competence initiation was followed using automated fluorescence time-lapse microscopy. We observed that competence synchronizes in neighboring microcolonies that are not in direct physical contact with each other (Fig. 3a, b and Supplementary Movie 1). In addition, more than 95% of the population became competent in sharp contrast with competence development in other species such as *Bacillus subtilis* where < 20% of the population enters the competent state[30, 31]. Both observations are consistent with the view that CSP diffuses extracellularly and drives competence development across the population. We provide experimental evidence for this claim by studying mixed

populations of our wild-type D39 strain and a *comC*⁻ mutant, which is unable to produce and export CSP, and therefore only develops competence in the presence of external CSP. We followed competence development in these mixed populations and found that competence can propagate from the wild type to the *comC*⁻ mutant without the need of cell-to-cell contact (Fig. 3c and Supplementary Movie 2). Note that the *comC*⁻ mutant does not activate competence when grown alone (Supplementary Fig. 4 and Supplementary Movies 3 and 4). This implies that CSP can diffuse extracellularly which confirms our previous finding in liquid culture that CSP can be recovered from the supernatant of a competent culture (Supplementary Fig. 1). It is worth noting that although cells release CSP to an extracellular pool, a fraction might remain attached to them or in close proximity due to diffusivity on the polyacrylamide surface. This can introduce variation in the microenvironment that different cells experience and therefore in the extent of synchronization of competence initiation.

In the experiments discussed so far, we used an encapsulated strain, in contrast to the studies of Claverys et al.[16] and Prudhomme et al.[20], which were mainly based on unencapsulated strains. We explored whether this could explain our different observations regarding the effect of inoculation density on competence development by studying additional strains: an unencapsulated version of our strain D39, the clinical isolate PMEN14 together with its unencapsulated version and *Streptococcus mitis*, which is naturally unencapsulated. Although we still observe that competence develops later for smaller inoculums, the slope of the RLU signal decreases with inoculation density for the two capsule knockouts and *S. mitis*, in agreement with the results of Claverys et al.[16] and Prudhomme et al.[20] (Supplementary Fig. 5). A decreasing slope of the RLU signal likely indicates decreased synchronization in competence development at lower inoculation density. Since cells are synchronized by the common extracellular CSP pool, a possible

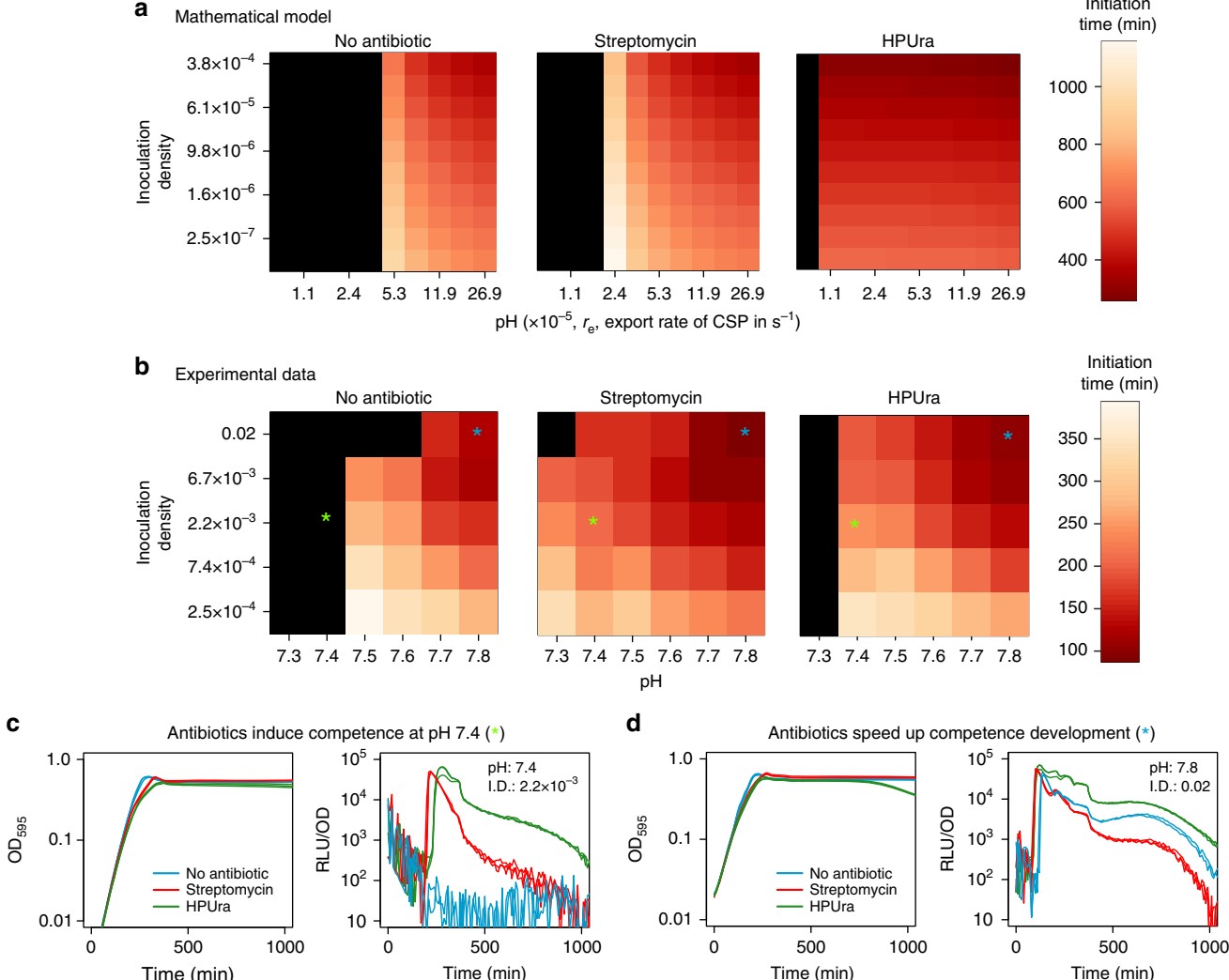

**Fig. 5** Competence is simultaneously regulated by cell density, pH and antibiotic stress. **a**, **b** Predictions of the mathematical model (**a**) and experimental data (**b**) on the dependency of the time of competence initiation on inoculation density, initial pH, and antibiotic stress. The x-axis in **a** corresponds to the rate of CSP export in the model, $r_e$, which is a proxy for pH. The *color scales* with the time of competence initiation with more intense *red* corresponding to faster development of competence. *Black* represents no competence development. In **b**, each box corresponds to the average initiation time of three replicates. Both the model and the experimental data show that competence develops faster at higher pH and higher inoculation densities. **c** Antibiotics induce competence at pH values that repress natural competence development. **d** At pH values that are not repressive for competence development (pH > 7.4), competence develops faster in the presence of antibiotics. The stars indicate which conditions are plotted in **c** and **d**. The concentrations of streptomycin and HPUra are 3 μg mL⁻¹ and 0.075 μg mL⁻¹, respectively. We chose these sub-MIC concentrations to minimize the effect of antibiotic stress on growth

scenario is that the absence of a capsule impedes cells from exporting all the CSP they produce. This would translate into less synchronization especially at low inoculation densities. In the extreme scenario where cells would not share any CSP with other cells, competence regulation would be in fact independent from cell density. Note, however, that for the capsule knockout of our strain we can still detect CSP in the cell-free supernatant and also observe competence propagation in the absence of cell–cell contact (Supplementary Fig. 6 and Supplementary Movie 5).

**pH and competence development.** In order to understand how environmental factors affect competence, we quantified the effect of external pH on natural competence development. We studied competence at a fine-grained range of pH values from 6.8 to 8.5 and found a clear-cut value that separated permissive from non-permissive external pH values for natural competence

development as reported before[13, 28]. For our media this was pH 7.4 (Fig. 4a). However, not only competence always developed at pH higher than 7.4 but the critical cell density for competence initiation decreased with increasing pH (Fig. 4a, b). Therefore, pH does not relate to competence as a binary permissive/non-permissive condition but competence development is more efficient in more alkaline media. The data suggests that for non-permissive pH conditions the cell density at which competence would initiate is above the carrying capacity of the medium, which was also previously proposed by Chen and Morrison[13].

To study the effect of pH on competence we deleted *comC* from the *comCDE* operon and put it under the control of an IPTG-inducible promoter[32] at an ectopic locus. We then tested competence development in this strain at different pH values (7.2, 7.4, 7.6) and at varying IPTG concentrations. For any given IPTG concentration, competence always developed earlier at higher pH. This indicates that cells need to express more *comC* to

reach the critical CSP threshold for competence activation the lower the pH (Supplementary Fig. 8a). In fact, at low IPTG concentrations, competence only develops above pH 7.4. The same pattern is observed in an IPTG-inducible *comCDE* strain: for a fixed level of *comCDE* expression, the time of competence initiation decreases with the pH (Supplementary Fig. 8b). Remarkably, in this genetic background competence hardly develops at pH 7.2 even when *comCDE* is fully induced. Cells might need to express more *comC* at lower pH because CSP export and/or detection reduces with decreasing pH. This would also explain why competence barely develops at low pH in the IPTG-inducible *comCDE* strain since for a fixed level of *comE* expression, reduced CSP export and/or detection would bias the ComE~P/ComE ratio towards no competence development. We tested whether pH affects CSP detection by using a *comC⁻* mutant. We performed experiments with medium at different initial pH values where we added various concentrations of synthetic CSP, using the *comC⁻* mutant. We found that competence was mainly dependent on the CSP concentration and only minor differences were found among media with different pH (Supplementary Fig. 8c). This suggests that competence development is not mainly mediated by pH-dependent CSP detection so possibly it is mediated by pH-dependent export. As peptidase-containing ATP-binding cassette transporters such as ComAB require ATP to transport substrates[33], it might be that the proton motive force influences its activity. Therefore, we incorporated the effect of pH in our model by changing the rate at which cells export CSP. In agreement with the experimental results, the modified model confirms that the density of competence initiation decreases with the rate of export of CSP and thus with higher pH. Also, the model predicts that for rates of CSP export below a certain threshold competence does not develop any more since cells never manage to accumulate enough CSP for competence to initiate (Fig. 4c). Note however that this is a simplification of the effect of pH in competence regulation since pH might also affect ComD and/or the stability of CSP (as in other QS systems[6]) and as our data suggest it might also be involved in the shutdown of competence (Supplementary Fig. 8c). However, regardless of the exact mechanism, as long as higher pH increases the rate at which single cells produce and/or sense CSP, our model predicts that the density at which the critical CSP concentration for competence activation is attained will decrease with increasing pH (see Section C of Supplementary Note 1 for a simple mathematical argument).

Finally, we assessed the joint effect of pH and cell density on competence regulation. We did this by studying competence initiation for cultures inoculated at different cell densities in media with different pH both experimentally and using the model. The model predicts that competence will initiate earlier both for higher inoculation densities and more alkaline pH (Fig. 5a, *left panel*): while higher inoculation densities mean that more cells will start producing CSP after inoculation, higher pH increases the rate at which individual cells produce CSP. The experimental data is consistent with this prediction (Fig. 5b, *left panel*). Therefore, the observation that pH affects competence development is not conflicting with regulation by cell density because the CSP concentration depends on both of these factors.

**Induction of competence by antibiotics**. The induction of pneumococcal competence is affected by the presence of certain classes of antibiotics[15, 28], which has been considered additional evidence for the hypothesis that competence can be regulated independently of cell density[15, 16]. We evaluated this claim by studying the role of HPUra and streptomycin on competence regulation. We chose these antibiotics since the mechanisms by which they induce competence at the molecular level have been elucidated to some extent: HPUra stalls replication forks during DNA replication while initiation of DNA replication continues, thereby increasing the copy number of genes near the origin of replication (*oriC*). As a consequence, it upregulates transcription of *comAB, comCDE*, and *comX* as these operons are located proximal to *oriC*[28]. Streptomycin causes mistranslation and is thought to regulate competence via the membrane protease HtrA which targets misfolded proteins and also represses competence possibly by degrading CSP[34] (but see Supplementary Fig. 9). By increasing the amount of misfolded proteins, streptomycin could reduce the rate at which CSP is degraded by HtrA leading to competence induction.

We reproduced the effect of HPUra and streptomycin on competence regulation in our model by increasing the transcription rate of *comAB, comCDE*, and *comX* and by reducing the rate at which CSP degrades, respectively. Our model predicts that the presence of antibiotics lowers the pH threshold for competence development (Fig. 5a), since antibiotics can counteract the effect of acidic pH to the point that cells can still accumulate enough CSP to become competent. They do this by increasing the rate at which single cells produce CSP (reducing the number of cells needed to reach the critical CSP concentration for competence initiation) or by increasing the rate at which they sense CSP (reducing the critical CSP concentration for competence initiation). Also, it predicts that for pH values where competence is already induced without antibiotics, it will develop faster in the presence of antibiotics (Fig. 5a). We tested these predictions experimentally using antibiotics at concentrations that have a minimal effect on growth since we did not incorporate growth reduction due to antibiotic stress in the model. In agreement with previous studies[15, 28, 34] and with the model predictions, we find that antibiotics can induce competence at pH values that are repressive for natural competence development (Fig. 5b, c). We also find support for the second prediction of the model since for permissive pH values for natural competence development (above 7.4), competence is induced earlier in the presence of antibiotics (Fig. 5b, d). Remarkably, both the model and the experiments show that the combined effect of pH and cell density in the presence of antibiotics remains the same as when no antibiotics are added (compare *left panel* with *middle* and *right panels* in Fig. 5a, b): competence induction still occurs earlier for high densities of inoculation and more alkaline pH values. In the case of Streptomycin at pH 7.3 it is even possible to see that competence does not develop for the highest inoculation density as the population probably reaches carrying capacity before enough CSP is produced. Finally, note that there might be alternative mechanisms from the ones incorporated in the model by which antibiotic stress affects competence. For instance, antibiotics reduce growth and can induce stress responses that lead to global changes on transcription and translation.

**Bistable region for competence development and cell history**. An important feature of the competence regulatory network is the presence of a positive feedback that couples CSP detection to CSP production (Fig. 1). Signaling systems that contain positive feedback loops often exhibit switch-like responses resulting in the occurrence of alternative stable states[35]. We varied the strength of the positive feedback loop in the model by changing the rate of CSP export and found that the competence regulatory network exhibits bistability for a range of intermediate CSP export rates. In this range, the model predicts the existence of two alternative states where competence switches "ON" or "OFF" depending on the initial conditions (Fig. 6a).

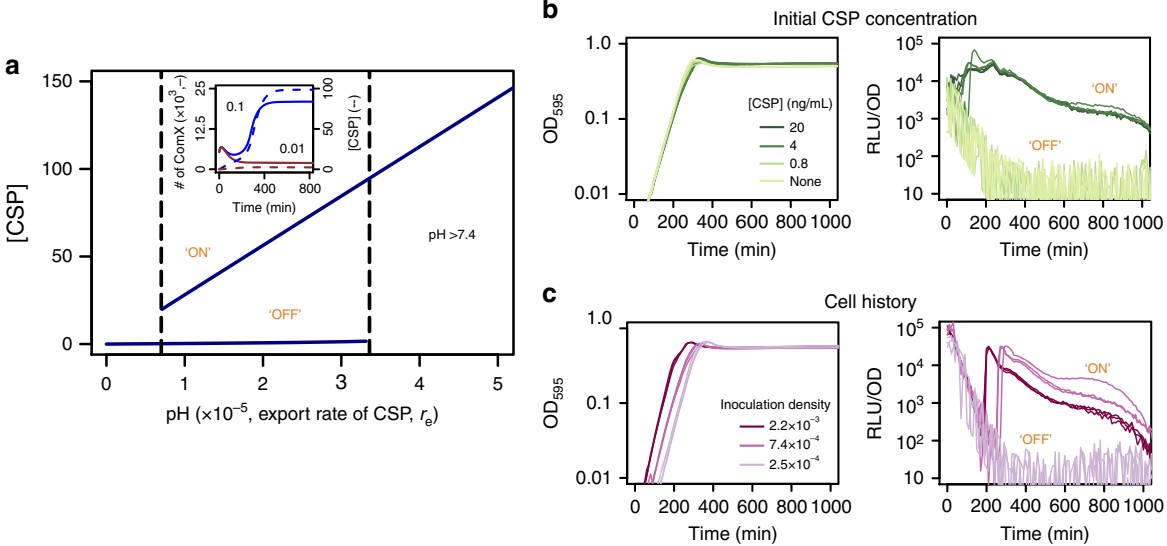

**Fig. 6** A bistable regime for competence development. **a** Extracellular concentration of CSP in response to the rate of CSP export. The model predicts the existence of a region where competence always switches on regardless of the initial conditions (which would correspond to pH > 7.4) and of a bistable region (bordered by the *dashed lines*). In the latter, the initial conditions can either switch on or not CSP production and subsequently competence development. (inset) In particular, the model predicts that in this region non-acid cell history can allow competence development if enough cells are inoculated since they can produce enough CSP to remain competent. The simulated inoculation densities are 0.1 (*blue*) and 0.01 (*brown*) and both the number of ComX molecules per cell (*solid line*) and the CSP concentration (*dashed line*) are shown. **b** Growth curves and competence expression measured as RLU units normalized by density for cells coming from acid preculture (pH 6.8) and inoculated in medium at pH 7.4 with different initial concentrations of CSP. Three replicates are shown per treatment and all the cultures are inoculated at $OD_{595}$ 0.002. **c** Growth curves and competence expression measured as RLU units normalized by density for cells coming from non-acid preculture (pH 7.9) and inoculated in medium at pH 7.4 at different initial densities. Three replicates are shown per inoculation density. Competence does not develop for cells inoculated at the same densities but coming from acid preculture (pH 6.8) (Fig. 5b, *left panel, second column*)

Since in the model the rate of CSP export is positively correlated to the pH, we expected to find a region of pH values exhibiting similar bistability as an additional experimental corroboration of the model. Indeed, we found support for the existence of a bistable region at pH 7.4 where the wild type developed competence if CSP was externally added in concentrations above $4 \, ng \, mL^{-1}$ (Fig. 6b). Thus, whereas competence always switched on for pH values above 7.4 regardless of the initial CSP concentration, for pH 7.4 both "ON" and "OFF" states were observed depending on the initial CSP concentration. Moreover, at pH 7.4 competence developed with $4 \, ng \, mL^{-1}$ of CSP a concentration that would not induce competence in the $comC^-$ mutant (Supplementary Fig. 8c), which indicates that CSP production in the wild type was kick-started by the initial addition of CSP resulting in enough overall CSP for competence induction.

Bistable systems usually exhibit hysteresis. For this reason, we expected that at pH 7.4 where both the "ON" and "OFF" states are attainable, cell history, would influence competence induction. From our previous experiments we determined that cells coming from acid preculture inoculated at pH 7.4 do not develop competence at any density of inoculation (Fig. 5b, *left panel, second column*). We then studied whether there is history dependence by inoculating cells coming from non-acid preculture at pH 7.4. We found that cells coming from a non-acid preculture became competent when inoculated at densities of $OD_{595 \, nm}$ $7.4 \times 10^{-4}$ or higher, which demonstrates that cell history can influence competence development at this pH (Fig. 6c). Past history has an effect on competence because it determines the state of the machinery for CSP production, which is "OFF" when cells come from acid preculture but "ON" when they come from non-acid conditions. This explains why the effect of non-acid cell history appears from a minimum inoculation

density, since enough cells need to be inoculated in order for them to produce the amount of CSP necessary for the system to remain "ON" (Fig. 6a, inset). We then hypothesized that at pH 7.3 the critical inoculation density of cells coming from non-acid history would have to be even higher than the one at pH 7.4 as the model predicted that a higher initial concentration of CSP would be necessary for the system to remain "ON" at lower pH. We confirmed this prediction experimentally by showing that at pH 7.3 competence does not develop for an inoculation density of $OD_{595 \, nm}$ $7.4 \times 10^{-4}$ (as for pH 7.4) but from $2.2 \times 10^{-3}$ upwards (Supplementary Fig. 10). Thus, our results show that, as a consequence of the positive feedback involved in CSP production, previous exposure to different environmental conditions can determine whether competence is induced or not by modifying the state of the machinery for CSP production and/or sensing.

## Discussion

Recently, the view that bacteria use autoinducers as QS signals has been debated since autoinducer concentration can change in response to the environment. Here, we show experimentally that cell density, pH and antibiotic stress simultaneously regulate competence development in *S. pneumoniae* (Figs. 2–5), a system classically framed in the paradigm of QS. Using a mathematical model, we show that this occurs because pH and antibiotics modify the rates at which single cells produce and sense CSP and therefore the strength of the positive feedback loop coupling CSP detection to CSP production (Figs. 4 and 5). This environmental dependency does not override regulation by cell density but rather modulates the relationship between the number of cells and the CSP concentration. A fundamental aspect to the dependency on cell density is that cells share CSP with others. Importantly, here we provide evidence both in

liquid culture (Supplementary Fig. 1) and through single-cell observations (Fig. 3) that CSP is exported to the extracellular space. Finally, we show that competence development is history-dependent since past environmental conditions can modify the status of the machinery to produce and respond to CSP determining whether competence switches on or not (Fig. 6). Hysteresis in the competence response might be especially important in the natural niche of the pneumococcus, the human nasopharynx. In particular it is consistent with the observation that there is constitutive upregulation of competence in pneumococcal biofilms during nasopharyngeal colonization[36]. In this context, once competence is triggered for the first time, cells would be primed to rapidly initiate another round of competence.

Why is competence controlled by CSP? CSP does not act as a timing device in our encapsulated strain since competence develops in a cell-density-dependent manner without the necessity of cell–cell contact[16, 17, 20] (Figs. 2 and 3). Regarding the hypothesis that CSP is a probe to test diffusion[21], our results suggest that focusing on diffusion alone oversimplifies the information and functionality that cells can gather through CSP production. We hypothesize that by using an autoinducer peptide, bacteria can coordinate the development of competence and in particular the expression of fratricins and bacteriocins, which are under the control of the competent state. These proteins can lyse or inhibit the growth of surrounding cells that are not competent, increasing the efficiency of genetic transformation and mediating competition with other bacteria[37–40]. By coordinating competence expression via CSP, an isogenic bacterial population can increase the total concentration of secreted fratricins and bacteriocins in times where population density is high, which likely translates into a higher amount of lysed cells and therefore potential DNA donors. Importantly, coordinating competence expression can also prevent the killing of clonal siblings since immunity to these proteins comes with the competent state. Note however that the extent to which cells synchronize competence development may vary depending on the strain, genotype and growth conditions. In particular, our results suggest that unencapsulated strains may synchronize less, which would explain the difference between the findings reported by Prudhomme et al.[20] and our study. Decreased synchronization may result from cells exporting less CSP to the extracellular space and keeping more to themselves. In fact, in other species like Streptococcus thermophilus where competence is controlled by ComS, a peptide that rapidly gets imported back into the cell, the rate of competence development decreases with the inoculation density[41] as observed for the unencapsulated pneumococcal strains. Finally, note that S. pneumoniae grows primarily in biofilms where there is heterogeneity in the physiological status and microenvironment that different cells experience. This can certainly influence the degree of synchronization in competence initiation across a population especially in the light of our findings that both current and past environmental conditions affect the competence regulatory network. Indeed, recent work in Streptococcus mutans showed heterogeneous competence activation of cells within biofilms and upon different environmental pH ranges[42, 43]. While studies of well-mixed cultures give insight into the response mechanism shared by all cells in a population, additional work is needed in the future to study how CSP production and detection by individual cells is shaped by their spatial context and history and to unravel how individual responses translate into patterns of population synchronization across different genotypes and strains.

What is the relevance of the information carried by CSP? Alkaline pH and antibiotic stress can induce competence by increasing the rate at which single cells produce and sense CSP.

We expect this to be a general mechanism by which sources of stress that are alleviated through competence induce this state (e.g., mobile genetic elements as hypothesized by Croucher et al.[44]). Upregulating competence in the presence of antibiotics can increase survival by activating the expression of stress response genes[7, 19], facilitating repair of damaged DNA and mediating acquisition of resistance[19, 45]. Our findings suggest that strategies to prevent competence development in response to antibiotics can focus on counteracting the effect of antibiotics on the rate at which cells produce or sense CSP. Regarding the benefits of upregulating competence with alkaline pH, these are less clear and could be an example of a non-adaptive response resulting from the inherent biochemical properties of ComAB and possibly ComD. Importantly, CSP can integrate additional environmental cues like oxygen availability through the CiaRH two-component system, which represses comC post-transcriptionally and is required for virulence expression and host colonization[12, 46]. In fact, CiaRH is key for the regulation of competence by multiple environmental signals in other streptococci like S. mutans[47].

Our findings support the view that functional hypotheses stressing individual factors like diffusion or population density underplay the complexity of information integrated by QS systems[4, 48–52]. Although the term "quorum sensing" overemphasizes the role of population density, we advocate for keeping it due to its widespread use and the fact that density will modify autoinducer concentration in any autoinducer production system. Crucially, QS should be used in a broad sense acknowledging that bacteria integrate past and current environmental factors in addition to population density into their QS responses. This view might be very useful for other autoinducer production systems like competence in Vibrio cholerae, where the synthesis of the autoinducer, CAI-1, depends on the intracellular levels of cAMP–CRP and therefore might incorporate information on the metabolic status of the cell[53, 54]. Also in other systems, clear links between signal production, quorum threshold and environmental conditions have been shown to affect QS[55–59].

Given that many biotic and abiotic factors can modify autoinducer concentrations[60], future work should aim to study the relevance of such factors in the natural context where bacteria secrete autoinducers. Such work is crucial to assess whether upregulating QS in response to a particular factor provides a benefit for bacteria or is merely a result of the biochemical properties of the QS regulatory network. An interesting possibility is that, as in other biological systems[61], bacteria could perform collective sensing of the environment through social interactions. In this context, by secreting autoinducers cells could share individual estimates of environmental conditions (e.g., antibiotic stress) for which upregulating QS is beneficial. Then, autoinducer secretion would provide a way to get a more reliable estimate of the environmental conditions by allowing a population to pool estimates made by individual cells. Importantly, such a role for autoinducer secretion would explain the dependency of QS on both cell density and the environment.

## Methods

**Bacterial strains and growth conditions**. All pneumococcal strains used in this study are derivatives of the clinical isolate S. pneumoniae D39[27] unless specified otherwise. To monitor competence development, strains either contain a transcriptional fusion of the firefly luc and the gfp gene with the late competence gene ssbB or a full translational ssbB–gfp fusion. Cells were grown in C+Y complex medium at 37 °C. C+Y was adapted from Adams and Roe[62] and contained the following components: adenosine (65 µm), uridine (107 µM), L-asparagine (331 µM), L-cysteine (71 µM), L-glutamine (150 µM), L-tryptophan (29.4) µM, casein hydrolysate (5 g L$^{-1}$), BSA (8 mg L$^{-1}$), biotin (2.46 µM), nicotinic acid (4.87 µM), pyridoxine (3.4 µM), calcium pantothenate (5.04 µM), thiamin (1.9 µM), riboflavin

(0.744 μM), choline (48 μM), CaCl$_2$ (113 μM), K$_2$HPO$_4$ (48.8 mM), MgCl$_2$ (2.46 mM), FeSO$_4$ (1.8 μM), CuSO$_4$ (2 μM), ZnSO$_4$ (1.74 μM), MnCl$_2$ (40 μM), glucose (11.1 mM), sodium pyruvate (2.7 mM), saccharose (944 μM), sodium acetate (24.4 mM), and yeast extract (2.5 g L$^{-1}$).

**Construction of recombinant strains.** To transform *S. pneumoniae*, cells were grown in C+Y medium (pH 6.8) at 37 °C to an OD$_{595}$ of 0.1. Then, cells were treated for 12 min at 37 °C with synthetic CSP-1 (100 ng mL$^{-1}$) and incubated for 20 min at 30 °C with the transforming DNA. After incubation with the transforming DNA, cells were grown in C+Y medium (pH 6.8) at 37 °C for 90 min. *S. pneumoniae* transformants were selected by plating inside Columbia blood agar supplemented with 3% of defibrinated sheep blood (Johnny Rottier, Kloosterzande, The Netherlands) and the appropriate antibiotics. Working stocks of cells were prepared by growing cells in C+Y (pH 6.8) until an OD$_{595}$ of 0.4. Cells were collected by centrifugation (1595 × *g* for 10 min) and resuspended in fresh C+Y medium (pH 6.8) with 15% glycerol and stored at −80 °C. See the details on the construction of the strains below and see Supplementary Table 1 for a list of all the strains used in this study.

*Strain DSM2*: To follow competence development during pneumococcal growth, a transcriptional fusion of two reporter genes, *luc* (firefly luciferase) and *gfp*, to the late competence promoter P$_{ssbB}$ was used. The plasmid pLA18[28], containing the P$_{ssbB}$–*luc*–*gfp* construct was transformed into *S. pneumoniae* D-PEP1[63] and transformants were selected on Columbia blood agar supplemented with 1 μg mL$^{-1}$ tetracycline. The P$_{ssbB}$–*luc*–*gfp* construct integrates into the *bgaA* locus, and correct transformants were verified by PCR.

*Strains ADP243 and ADP62*: To prevent the production of CSP, the original *comC* gene was replaced by an erythromycin resistance marker, leaving the promoter and therefore the polycistronic nature of *comDE* intact. The upstream region was amplified using primers LAG11 (GGCGGATCCGGCAGTTTGTGTA ATAGTAC) and ADP2/48+AscI (ACGTGGCGCGCGCCGTTCCAATTTAACTGTG TTTTTCAT), the downstream region with primers ADP2/47+BamHI (ACGTGG ATCCGAAATAAGGGGAAAGAGTAATGGATTTATTTG) and LAG54 (AATC GCCATCTTCCAATCCC), and the erythromycin resistance marker with 0292_ery_R+BamHI (GCGGATCCTGTCTTTGACCCAATCATTC) and sPG19 +AscI (ACGTGGCGCGCCCGGAGGAATTTTCATATGAAC). All three fragments were digested with the proper restriction enzymes (AscI and/or BamHI) and ligated. The Δ*comC::ery* fragment containing the erythromycin resistance marker flanked by the sequence up- and downstream of *comC* without altering the natural transcription of *comDE*⁻ was transformed into D39 and DLA3[28] resulting in strains ADP243 (Δ*comC::ery*) and ADP62 (Δ*bgaA*:: P$_{ssbB}$–*luc*, Δ*comC::ery*), respectively. Transformants were selected on Columbia blood agar containing 0.25 μg mL$^{-1}$ erythromycin. Correct deletion of *comC* was verified by PCR and sequencing.

*Strains ADP49 and ADP51*: To monitor natural competence in two clinical isolates, the PCR product of the fragment *ssbB–luc–kan* from strain MK134[28] was transformed into the pneumococcal strain PMEN14 and the *S. mitis* strain NTCC10712. Transformants were selected on Columbia blood agar containing 250 μg mL$^{-1}$ kanamycin resulting in strains ADP49 (PMEN14, *ssbB–luc*) and ADP51 (*S. mitis ssbB–luc*), respectively.

*Unencapsulated strains ADP25, ADP26, and ADP92 and ADP238*: To delete the polysaccharide capsule, the whole *cps* operon was replaced with a chloramphenicol resistance cassette. The primers used were: ADP1/51 (CGGTCTTCAGTATCAGG AAGGTCAG) and ADP1/52+AscI (CGATGGCGCGCCCTTCTTTCTCCTTAAT AGTGG) to amplify the upstream region; primers ADP1/53+NotI (CACGGCGG CCGCGAGAAAGTTTTAAAGGAGAAAATG) and ADP1/54 (GATAGAGACGA GCTGCTGTAAGGC) to amplify the downstream region; sPG11+AscI (ACGTGG CGCGCCAGGACGCATATCAAATGAAC) and sPG12+NotI (ACGTGCGGCCG CTTATAAAAGCCAGTCATTAG) to amplify the chloramphenicol cassette. Products were digested with the appropriate enzymes (AscI and/or NotI) and ligated. Strains D39, DLA3, and ADP49 were transformed with the ligation product and transformants were selected on Columbia blood agar containing 4.5 μg mL$^{-1}$ chloramphenicol resulting in strains ADP25 (Δ*cps::chl*), ADP26 (Δ*bgaA*:: P$_{ssbB}$–*luc*, Δ*cps::chl*), and ADP92 (PMEN14, *ssbB–luc*, Δ*cps::chl*). To prevent the production of CSP in the unencapsulated D39, the Δ*comC::ery* fragment was transformed into ADP25, resulting in strain ADP238. Transformants were selected on Columbia blood agar containing 0.25 μg mL$^{-1}$ erythromycin. Correct deletion of *comC* was verified by PCR and sequencing.

*Strains ADP249 and ADP151*: To follow competence development of microcolonies during time-lapse microscopy, the PCR product of the C-terminal fusion of *ssbB* with *gfp* and a downstream kanamycin cassette were amplified using chromosomal DNA of strain RA42 as a template[64] and transformed into the D39 strain and into its unencapsulated variant ADP25 resulting in strains ADP249 (P$_{ssbB}$–*ssbB–gfp*) and ADP151 (P$_{ssbB}$–*ssbB–gfp*, Δ*cps::chl*), respectively. Transformants were selected on Columbia blood agar containing 250 μg mL$^{-1}$ kanamycin.

*Strains ADP247 and ADP248*: Plasmid pPEP43[63] containing a constitutive promoter driving the expression of the red fluorescent protein mKate2 (cytoplasmic localization) was transformed into strain D39 and its unencapsulated derivative ADP25. pPEP43 integrates into the *cep*-locus, and transformants were

selected on Columbia blood agar containing 125 μg mL$^{-1}$ spectinomycin. The resulting strains, ADP235 (Δ*cep::p3-mkate2*) and ADP244 (Δ*cep::p3-mkate2*, Δ*cps::chl*), were transformed with the PCR product P$_{ssbB}$–*ssbB*–*gfp* including the kanamycin cassette (same as for strain ADP249). Transformants were selected on Columbia blood agar containing 250 μg mL$^{-1}$ kanamycin. Finally, the resulting strains, ADP245 (Δ*cep::p3-mkate2*, P$_{ssbB}$–*ssbB–gfp*) and its unencapsulated variant ADP246 (Δ*cep::p3-mkate2*, Δ*cps::chl*, P$_{ssbB}$–*ssbB–gfp*), were transformed with the *comC::ery* fragment described above. This resulted in strains ADP247 (Δ*cep::p3-mkate2*, P$_{ssbB}$–*ssbB–gfp*, Δ*comC::ery*) and ADP248 (Δ*cep::p3-mkate2*, Δ*cps::chl*, P$_{ssbB}$–*ssbB–gfp*, Δ*comC::ery*). Transformants were selected on Columbia blood agar containing 0.25 μg mL$^{-1}$ erythromycin.

*Strain ADP95*: To use an IPTG-inducible system, we first transformed the codon optimized *lacI* gene[32] into strain DLA3[28]. For that, we PCR-ed the fragment including *lacI* integrated into the *prsA*-locus together with a gentamycin resistance cassette from chromosomal DNA of strain DCI23[32] using primers OLI40 (CCATGGCATCAGCGAGAAGGTGATAC) and OLI41 (GCGGC CGCAGGATAGAAAGGCGAGAG) and transformed the resulting PCR product to strain DLA3 while selecting for gentamycin (20 μg mL$^{-1}$) resulting in strain ADP95 (D39, Δ*bgaA*::P$_{ssbB}$–*luc*, Δ *prsA*:*lacI*).

*Strains ADP112 and ADP107*: To control the production of *comC*, and thereby CSP, a strain was constructed with an ectopic copy of *comC* (at the *cep* locus) under control of the IPTG-inducible promoter P$_{lac}$[32], and the subsequent deletion of the original *comC* (replaced by an erythromycin resistance cassette). The inducible system was created using BglFusion cloning[63]. To amplify the *comC* fragment, primers ADP2/38 (CAGTGGATCCGGTTTTTGTAAGTTAGCTTACAAG) and ADP3/31 (CAGTCTCGAGCCCAAATCCAAATAAATCCATTAC) were used using D39 chromosomal DNA as a template. To control the production of *comCDE*, a strain with an IPTG-inducible *comCDE* was created, with the deletion of the original *comCDE* locus. The inducible system was created as explained above. To amplify the *comCDE* fragment, primers ADP2/38 (CAGTGGATCCGG TTTTTGTAAGTTAGCTTACAAG) and ADP2/58 (ACGTCTCGAGGCGGGCCG CCAATTTCTTGCTAATTGTC) were used using chromosomal DNA of D39 as a template. PCR products were digested with BglII and XhoI and were ligated with similarly digested pPEP1 plasmid containing the promoter P$_{lac}$[32]. The ligations were transformed into strain ADP95 and transformants were selected on Columbia blood agar containing 125 μg mL$^{-1}$ spectinomycin. To these strains, either the PCR product Δ*comC::ery* (see above) or a PCR product containing Δ*comCDE::chl* from strain MK135[28] was transformed and transformants were selected on Columbia blood agar containing either 0.25 μg mL$^{-1}$ erythromycin or 4.5 μg mL$^{-1}$ chloramphenicol resulting in strains ADP112 (Δ*bgaA*::P$_{ssbB}$–*luc*, Δ*cep*:P$_{lac}$-*comC*, Δ*comC::ery*) and ADP107 (Δ*bgaA*::P$_{ssbB}$–*luc*, Δ*cep*:P$_{lac}$-*comCDE*, Δ*comCDE::chl*), respectively.

*Strain MK356*: To construct MK356 (Δ*bgaA*:: P$_{ssbB}$–*luc–gfp*, Δ*htrA::ery*), *htrA* was replaced with an erythromycin resistance gene. A region upstream of *htrA* was amplified from genomic DNA of *S. pneumoniae* D39 using primers htrA-up-F (5′-GAACCTGCGACCGTTCGCTTAGAAGG-3′) and htrA-up-R+BamHI (5′-GCGC GGATCCTCCATATGTTTGAATTACTG-3′) and a region downstream of *htrA* was amplified with primers htrA-Dwn-F-EcoRI (5′-CGCGGAATTCGACATCTA TGTAAAGAAAGC-3′) and htrA-Dwn-R (5′-GCTGTTGATAATTCTACTATAT TCTTC-3′). An erythromycin resistance gene was amplified from plasmid pORI28[65] using primers EmR-F+BamHI (5′-GCGCGGATCCTATGAACGAGAA AAATATAAAACAC-3′) and EmR-R+EcoRI (5′-CGCGGAATTCGCAGTTTATG CATCCCTTAACTTAC-3′). The three fragments were digested with appropriate restriction enzymes (BamHI and/or EcoRI) and ligated. The Δ*htrA::ery* fragment, containing the erythromycin resistance gene flanked by the sequence up- and downstream of *htrA*, was transformed into strain DSM2. Transformants were selected on Columbia blood agar containing 0.25 μg mL$^{-1}$ erythromycin. Correct deletion of *htrA* was verified by PCR.

**Density and luminescence assays.** Cells were pre-cultured in acid C+Y (pH 6.8) or in non-acid C+Y (pH 7.9) at 37 °C to an OD$_{595\,nm}$ of 0.1. Right before inoculation, they were collected by centrifugation (8000 rpm for 3 min) and resuspended in fresh C+Y (pH 7.9). Unless indicated otherwise all experiments were started with an inoculation density of OD$_{595}$ 0.002 with cells from an acid preculture. Luciferase assays were performed in 96-wells plates with a Tecan Infinite 200 PRO luminometer at 37 °C as described before[63]. Luciferin was added at a concentration of 0.5 mg mL$^{-1}$ to monitor competence by means of luciferase activity. Optical density (OD$_{595\,nm}$) and luminescence (relative luminescence units (RLU)) were measured every 10 min. The time and density of competence initiation correspond to the first time point where the RLU signal is equal or above 200 units. RLU is used instead of RLU/OD for Fig. 2 because (1) when competence develops the rate at which the RLU signal increases is faster than the growth rate and (2) due to the very low inoculation densities used for Fig. 2 the RLU/OD can be very high at the start (clearly before competence has developed). The value of 200 units was chosen because once this value is reached competence always developed. The effect of pH on competence development was studied by inoculating cells in C+Y at a range of pH values from 6.8 to 8.5. pH was adjusted by adding HCl and NaOH. The effect of antibiotics was studied by adding streptomycin (3 μg mL$^{-1}$) and HPUra (0.075 μg mL$^{-1}$) to C+Y.

**Detection of CSP in cell-free supernatant**. *S. pneumoniae* D39 wild-type and its *comC*-deficient version, ADP243, were grown in non-acid C+Y (pH 7.9) at 37 °C to an $OD_{595\,nm}$ of 0.1. Cells were spun down by centrifugation at $20,000 \times g$ for 5 min. The supernatant was sterilized by filtering twice through 0.2 μm filters. The supernatant was plated on Columbia blood agar to confirm that no viable cells were present. For both the wild-type and the *comC*-deficient strain, the cell-free supernatant was diluted 1:1 with 2× concentrated C+Y medium containing luciferin, and pH was adjusted to the indicated value by addition of HCl. The indicator strain DSM2 pre-grown in C+Y pH 6.8 was then inoculated, and growth and luciferase activity was monitored as described before.

**Time-lapse fluorescence microscopy**. A polyacrylamide slide was used as a semi-solid growth surface to spot the cells for time-lapse microscopy. This slide was prepared with C+Y (pH 7.9) and 10% acrylamide as reported previously[66]. Cells were pre-cultured in acid C+Y (pH 6.8) and right before inoculation on the slide they were resuspended in fresh C+Y (pH 7.9) as explained before. Phase contrast, GFP and RFP images were obtained using a Deltavision Elite microscope (GE Healthcare, USA). Time-lapse videos were recorded by taking images every min after inoculation unless specified otherwise. File conversions were done using Fiji and analysis of the resulting images was done using Oufti[67].

**Mathematical model**. A mathematical model of the network of competence regulation (Fig. 1) was developed as a system of ODEs. The model incorporates the protein interactions involved in sensing CSP via the two-component system formed by ComD and ComE and exporting it via ComC and ComAB. Additionally, it explicitly models the interaction of ComE and ComE~P with the gene promoters of *comAB*, *comCDE*, and *comX*. This is important since ComE~P binds these promoters as a dimer introducing non-linearity into the system, which underlies the observed bistability. Population growth is logistic and it is assumed that cells are homogeneous and that they export all CSP they produce to a common extracellular pool. See Supplementary Note 1 for the equations and further description.

**Data availability**. The authors declare that the data supporting the findings of the study are available in this article and its Supporting Information files, or from the corresponding authors upon request.

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

## Acknowledgements

We thank Martin Ackermann, Melanie Blokesch and two anonymous reviewers for helpful comments on an earlier version of this manuscript before submission to this journal, and Katrin Beilharz for the *htrA::ery* construct. S.M.-G. and G.S.v.D. were supported by Starting Independent Researcher Grant 309555 of the European Research Council and a VIDI fellowship (864.11.012) of the Netherlands Organization for Scientific Research (NWO). A.D. was supported by Marie Skłodowska-Curie fellowship 657546. Work in the Veening lab is supported by the EMBO Young Investigator Program, a VIDI fellowship (864.12.001) and ERC starting grant 337399-PneumoCell.

## Author contributions

S.M.-G., A.D., R.A.S., M.K., G.S.v.D. and J.-W.V. designed research; S.M.-G., A.D., R.A.S. and M.K. performed experiments; S.M.-G. and G.S.v.D. developed the model; S.M.-G., A.D., R.A.S., M.K., F.J.W., G.S.v.D. and J.-W.V. analyzed data; and S.M.G., F.J.W., G.S.v.D. and J.-W.V. wrote the paper.

## Additional information

**Competing interests:** The authors declare no competing financial interests.

