## [Peer Review File · Nature Communications]

Editorial Note: This document only contains reviewer comments and rebuttal letters for versions considered at Nature Communications. Mentions of other journals have been redacted.

Reviewers' comments:

Reviewer #1 (Remarks to the Author):

In this paper, Moreno-Gamez et al investigate effects of environmental cues, cell density and cell history on the quorum sensing system that regulates competence of *S. pneumoniae*. They demonstrate that the QS system integrates all these different factors and propose plausible mechanisms for this integration. Their interpretation of the experimental data is further supported by simulations of a mathematical model of the QS system. All-in-all, this is a clearly written and largely self-consistent study that will be interesting to a wide readership. Although the observation that the QS response is modulated by environmental cues is not particularly surprising, the novelty of this manuscript lies in providing the integral view of the interplay between multiple factors inducing the same system. I thus believe that this manuscript will be important to the field of microbial communication, helping to establish a more complex view of the QS-mediated regulation.

I have only two specific comments:

- 1) While it is absolutely clear that pH and antibiotic stress modulate the activity of the *ssbB* reporter, the authors should be more careful in making the statement that these factors directly affect the QS system. I am particularly critical about their statement in the Discussion (p. 16, lines 386-388) "As shown by our model, alkaline pH and antibiotic stress induce competence by increasing the rate at which single cells produce and sense CSP". Both effects have been explicitly incorporated into the model by the authors, so here the model only "shows" what is included into it. While I find the case for the pH-dependent regulation of CSP production reasonable (but not fully, as acknowledged by the authors themselves in the Results) supported by the experiments, the regulation of the CSP production by the antibiotic stress is to my understanding only a plausible hypothesis. Is there any reason to exclude the possibility that antibiotic stress regulates competence genes independent of the QS system?
- 2) Given the interplay between pH and CSP signaling, did the authors test the pH of the medium during growth?

Reviewer #2 (Remarks to the Author):

I have reviewed this paper for [redacted] before. The authors have addressed some but not all of my previous points in their revision. The study investigates quorum sensing gene expression in *Streptococcus pneumoniae*. Specifically, it considers the induction of competence by a secreted autoinducing peptide, CSP, in response to cell density and environmental conditions (medium pH). This is a thorough, well-written study that synthesizes, and attempts to resolve, seemingly discordant findings from previous studies. It alternates between mathematical model and experimental verification.

An initial criticism of mine was that the study's broader conclusions mainly affirm established principles of quorum sensing gene regulation, namely that (1) quorum sensing depends on a critical signal concentration rather than cell density per se, and (2) environmental conditions such as nutrient limitation and abiotic stressors can affect quorum sensing. The authors do a better job now of acknowledging these established principles in the Discussion. The authors also do a better job of pointing out inconsistencies in their specific field. In this context, the study refutes the role of CSP as a timing device, as suggested previously. I still think that the timing-device hypothesis is implausible in the first place and rests on questionable data, but I also appreciate that it nevertheless seems to have established itself in the streptococcal competence literature. This study therefore makes an important contribution to the field by directly addressing and rectifying it.

Other points:

I have suggested before that the authors should measure pH-dependent CSP production in *S. pneumoniae* to confirm mathematical modeling and mutant analysis (p 10-11). I still think this would be a good idea.

The idea of "cell history" is still not well defined and its physiological/ecological significance is not discussed. From a purely technical point of view, it is obvious that in the experimental system used, inoculation of batch cultures with fully induced cells will activate competence earlier than inoculation with uninduced cells.

The authors state that "little is known about why competence is controlled by an autoinducer peptide like CSP" (p 3, I 65). While this statement implies an evolutionary "purpose" that may be difficult to test, there is nevertheless a reasonable hypothesis worth stating here, namely that the concentration of extracellular DNA from lysed, non-competent cells increases with cell density. This seems so obvious that alternative ideas ("timing device", "diffusion sensing") should not completely distract from it and relegate its first mention to the Discussion (p 15).

Reviewer #3 (Remarks to the Author):

This manuscript consists of mathematical modeling of the signal transduction pathway that leads to late competence gene expression by *Streptococcus pneumoniae* coupled with a series of in vitro experiments that examine predictions of the model in the context of environmental inputs. The model appears carefully constructed and includes the required participants for induction of competence. Although the model omits other known inputs to make the analyses tractable, it is a valid starting point. The authors focus on the effects of pH, cell density and antibiotics on development of competence. These three external variables have already been established to influence competence signaling, although the mechanisms by which they do so are somewhat poorly understood. Two primary components of the presentation include experiments and interpretations that contradict prior studies that a) demonstrated that CSP may function as a timing device to coordinate competence activation and b) that suggest that CSP may first activate competence within a sub-population of cells, which in turn leads to propagation of competence signaling via CSP and activation of *com* gene expression across the entire population.

There is a considerable amount of data presented. The analyses using the mathematical models and the experiments that the authors present have been carefully designed and performed, the results are clearly presented, the paper is well written, and the interpretations they offer are reasonable - within the confines of their models and experimental design. There are, however, some deficiencies. There is no data presented to try to explain the differences in the present results and prior studies; for example, the use of different strains and growth conditions could account for the disparate behaviors in this and previous studies. The authors attempt to provide a unifying theory for the role of CSP to help resolve the debate about whether CSP is a true quorum sensing (QS) molecule by positing that CSP should be considered a QS signal, but only in the context of the cellular (micro)environment. Like some of the experimentation and other conclusions, this proposal is not entirely novel and does not substantially advance the field or create new concepts for QS and/or competence. There were only modest attempts at integrating the findings with the biology, physiology and pathogenesis of the organism, or with similar behaviors reported for related organisms. Some of the more novel and interesting findings, such as the effects of pH and potential bi-stable behavior elicited at pH 7.3, were not thoroughly investigated.

Specific Comments

Line 28. Probably should be modified to say that "competence induction can be simultaneously influenced by cell density..."

Line 34. Here and throughout, the concept of "cell history" is discussed. In this case, it refers to whether cells were cultured in conditions that allowed for prior activation of com gene expression. Subsequently, the kinetics of com activation is stated to depend on the "history" of the cells. It would probably be more clear and accurate to simply state that the authors are comparing behaviors of cells that are already actively transcribing com genes with cells in which com genes have not yet been induced. Not surprisingly, those that are already induced for com expression show more rapid activation of the pathway when transferred to inducing conditions. One possible analogy could be that if cells are grown on an alternative carbohydrate source (e.g. lactose), a dramatic decrease in the lag phase is observed when those cells are transferred to medium containing lactose, as compared to cells that were first cultured on glucose. In contrast, when one thinks of cell "history", it seems more common that this is referring to cells that have genetically and physiologically differentiated into a distinct state, such as persisters that arise in long-term culture. Thus, it seems more appropriate to replace the concept of the cells' "history" with using "previously induced" or "uninduced" cells in the experimental systems.

Line 50. There is considerable introductory material devoted to diffusion, which was not a variable directly tested in this study. Since diffusibility as a factor in intercellular communication has already been experimentally verified, and is intuitive, this section can be streamlined.

Line 59. Please clarify, is it the role of competence or QS that is the subject of debate?

Line 73 to end of paragraph. These studies, like the present one, use model systems that are dissimilar to what the organisms face in vivo. This is pointed out not to diminish the importance of any of the studies, but to emphasize that a) pneumococci in a biofilm are exposed to a variety of microenvironments and b) these biofilms contain cells in various stages of growth/growth arrest and are therefore quite different from the cells growing exponentially in planktonic culture. All of this work should therefore be clearly framed in the context of "under the conditions tested", since competence is exquisitely sensitive to microenvironments and the physiologic status of the cells. Carrying forward this concept throughout the paper, as well as including it when there is disagreement between two studies, would strengthen the presentation.

Line 114. The model that is constructed is fairly simple because the system itself is relatively simple, compared to some other organisms (e.g. Bacillus, certain other streptococci). Still, the authors should acknowledge that other inputs (e.g. ComW, factors that shut off competence) are present and explain how they may be accounted for in their model (e.g. ComX stability).

Line 152. Either at this point or in the discussion it would be appropriate to clarify that there are differences in the strain(s) and other variables that might account for the differences between results in citation 18 and this study. As written, it comes across as the authors inferring that the prior work is incorrect, without having done the necessary experiments to take such a position.

Line 162. This would be one example of the findings not being entirely novel, since this is a roundabout approach that reaffirms there is a dose-dependent response to CSP by pneumococci.

Line 172. It is not clear that there is a "common misconception" related to QS. In reality, most researchers would agree that if the environmental conditions could influence the real or effective concentration of the QS molecule, that you would begin to uncouple QS from cell density.

Line 188-204. Manipulating pH is one way to demonstrate this. Use of cells pre-treated with CSP could further support their hypothesis.

Figure 3. It would be helpful to see this data in a histogram-type display. The issue is whether

their data really do refute the idea of a sub-population or if there may be some issue with the methodology. If the model were modified a little, would the data appear more similar to that reported by Prudhomme et al. (Ref. 22)?

Figure 3. It seems a little surprising that *S. pneumo* could grow at pH 9.0, or even much above 8.0, yet according to 4a it does. If it cannot grow at pH 9.0, then one explanation for the data would be that, during the long lag period, the cells are producing acid and lowering the pH to a growth-permissive level. If that is the case, then the data should be reported and the conclusions modified accordingly.

Line 235. It is a little perplexing why a *comA* mutant was utilized instead of a *comC* deletion, it seems the latter would yield cleaner data. It also seems there should be consideration given to other explanations for why there are different behaviors of the WT and *comA* knockout other than rate of CSP export, even though the results are consistent with the mathematical model outcome.

Line 247. It appears the authors are trying to say that cells that are exposed to higher pH do not need to shunt as much ATP to maintaining the delta pH component of the pmf as do cells that are exposed to more acidic conditions; thus the rate of CSP export through ComAB can be higher because the former cells have greater ATP pools. This may or may not be true. While the authors make some concessions to other scenarios for altered CSP production below (line 249-53), it would be helpful to be more specific what they mean vis-à-vis impacts on pmf and the outcome of the experiments.

Line 257. It would strengthen the presentation and allay some of the concerns stated above if some experimental verification of this hypothesis were provided. That is, the authors could use MS to directly quantify CSP secreted by cells under different pH conditions, or use the supernatants derived from cells exposed to different pH values to see how effectively they activated a *com*-reporter strain (a surrogate measure of CSP concentration in the supernatant fluids).

The section related to antibiotic effects on competence should be tempered somewhat since the actual mechanisms by which these antibiotics are inducing competence are not well-defined. It cannot be excluded that factors related to growth arrest or overlap with other stress regulons could be modifying cell behavior independently of the variables being tested.

The induction of bi-stable behavior at pH 7.3 but not at 7.4 is perhaps the most intriguing and novel part of the story. While it is not unreasonable to suggest that the system is thresholded, allowing for potential stochastic behaviors, there is insufficient direct evidence. Use of other common technical approaches to prove the existence of responsive and non-responsive populations (e.g. microscopy, cell sorting) would strengthen the hypothesis.

Discussion. The three main issues surfacing in the discussion are related to the use of the concept of "history" of the cells, the tentative nature of the proof of bi-stable behavior, and a lack of citation of literature that is directly related to the study, but in which the work was conducted with organisms other than *S. pneumoniae*. There are a number of quality studies done on environmental effects on competence in related organisms that would be of value to include in the discussion. Likewise, it could be argued that any discussion of competence in streptococci should include some mention of the ComRS pathway.

Authors reply to the referees:

Reviewers' comments: Reviewer #1 (Remarks to the Author):

In this paper, Moreno-Gamez et al investigate effects of environmental cues, cell density and cell history on the quorum sensing system that regulates competence of *S. pneumoniae*. They demonstrate that the QS system integrates all these different factors and propose plausible mechanisms for this integration. Their interpretation of the experimental data is further supported by simulations of a mathematical model of the QS system. All-in-all, this is a clearly written and largely self-consistent study that will be interesting to a wide readership. Although the observation that the QS response is modulated by environmental cues is not particularly surprising, the novelty of this manuscript lies in providing the integral view of the interplay between multiple factors inducing the same system. I thus believe that this manuscript will be important to the field of microbial communication, helping to establish a more complex view of the QS-mediated regulation.

We thank the reviewer for the comments and support of the manuscript.

I have only two specific comments: 1) While it is absolutely clear that pH and antibiotic stress modulate the activity of the *ssbB* reporter, the authors should be more careful in making the statement that these factors directly affect the QS system. I am particularly critical about their statement in the Discussion (p. 16, lines 386- 388) “As shown by our model, alkaline pH and antibiotic stress induce competence by increasing the rate at which single cells produce and sense CSP”. Both effects have been explicitly incorporated into the model by the authors, so here the model only “shows” what is included into it. While I find the case for the pH-dependent regulation of CSP production reasonably (but not fully, as acknowledged by the authors themselves in the Results) supported by the experiments, the regulation of the CSP production by the antibiotic stress is to my understanding only a plausible hypothesis. Is there any reason to exclude the possibility that antibiotic stress regulates competence genes independent of the QS system?

We agree that the model does not provide any insight about the molecular mechanism by which pH and antibiotics induce competence. Instead, the model is useful to implement any environmental factor with a known mechanism and study its effect alone or in combination with other factors on competence induction (e.g. Figures 4C and 5A). As we argue in line 310 we chose the two antibiotics because their mechanism of action on the competence regulatory network is known. The regulation of CSP production and detection by HPUra and several other antibiotics has been shown in a previous paper from our group (Slager *et al.* 2014). This paper shows that by stalling replication elongation but not replication initiation, HPUra increases the copy number of genes near the

oriC, which includes *comAB* and *comCDE*. This thus results in an increase in the rates of CSP production and detection since *comC* is the precursor of CSP, *comAB* encodes the CSP export machinery and *comD* is the membrane-integrated histidine kinase that senses CSP. The authors showed that in a strain where *comCDE* is located far from the *oriC*, competence is not induced by HPUra anymore, which supports the idea that increasing the copy number of *comCDE* is key for induction of competence by HPUra. Regarding streptomycin, we incorporate this antibiotic into the model by lowering the rate of CSP degradation, a mechanism suggested by Stevens *et al.*, 2011. We acknowledge that these mechanisms are a simplification of the effect of antibiotic stress on competence. We now clearly state in the last part of the section on antibiotics that there might be additional ways in which antibiotics affect competence besides the mechanisms we incorporate.

“Finally, note that there might be alternative mechanisms from the ones incorporated in the model by which antibiotic stress affects competence. For instance, antibiotics reduce growth and can induce stress responses that lead to global changes on transcription and translation.”

We also added explanatory sentences in the text and in the caption of Figure 5:

“We tested these predictions experimentally using antibiotics at concentrations that have a minimal effect on growth since we did not incorporate growth reduction due to antibiotic stress in the model.”

“The concentrations of streptomycin and HPUra are $3 \mu\text{g mL}^{-1}$ and $0.075 \mu\text{g mL}^{-1}$, respectively. We chose these concentrations to minimize the effect of antibiotic stress on growth. ”

Finally, we replaced the sentence that the reviewer mentioned in the Discussion for:

“Alkaline pH and antibiotic stress can induce competence by increasing the rate at which single cells produce and sense CSP.”

2) Given the interplay between pH and CSP signaling, did the authors test the pH of the medium during growth?

We did test the pH of C+Y during growth and it decreases with cell density as now shown in new Figure S6. The pH decreases roughly linearly with cell density and it can go down for about 1.5 units once a culture has reached stationary phase. However, we think this effect is not very important for competence development in our conditions since the cell densities at which competence initiates are generally below 0.1 so acidification by cell growth is low. There are two scenarios where media acidification by growth can become more relevant. First, for high inoculation densities (>0.05), since in this case the medium gets

rapidly acidified after the first divisions. In fact, media acidification by growth might be another reason why competence does not develop when the inoculation density is 0.1 (Figure 2A). Second, when the initial pH of the medium is close to 7.4 and thus the density of competence initiation is above 0.1 (Figure 4B).

We have now included a sentence regarding this point in the legend of Figure 4:

“Note that the indicated pH is the initial pH of the medium, which does not stay constant due to acidification by growth (Figure S6). Although the pH drops considerably in fully-grown cultures, acidification is still minor at the density where competence develops.

Also, we now refer to initial pH in figure legends and captions to clarify this issue.

Reviewer #2 (Remarks to the Author):

I have reviewed this paper for [redacted] before. The authors have addressed some but not all of my previous points in their revision. The study investigates quorum sensing gene expression in *Streptococcus pneumoniae*. Specifically, it considers the induction of competence by a secreted autoinducing peptide, CSP, in response to cell density and environmental conditions (medium pH). This is a thorough, well-written study that synthesizes, and attempts to resolve, seemingly discordant findings from previous studies. It alternates between mathematical model and experimental verification. An initial criticism of mine was that the study’s broader conclusions mainly affirm established principles of quorum sensing gene regulation, namely that (1) quorum sensing depends on a critical signal concentration rather than cell density per se, and (2) environmental conditions such as nutrient limitation and abiotic stressors can affect quorum sensing. The authors do a better job now of acknowledging these established principles in the Discussion. The authors also do a better job of pointing out inconsistencies in their specific field. In this context, the study refutes the role of CSP as a timing device, as suggested previously. I still think that the timing-device hypothesis is implausible in the first place and rests on questionable data, but I also appreciate that it nevertheless seems to have established itself in the streptococcal competence literature. This study therefore makes an important contribution to the field by directly addressing and rectifying it.

We thank the reviewer for having a second look to our manuscript and giving additional feedback.

Other points:

I have suggested before that the authors should measure pH-dependent CSP production in *S. pneumoniae* to confirm mathematical modeling and mutant analysis (p 10-11). I still think this would be a good idea.

We have now included a more thorough mutant analysis by placing *comC* and *comCDE* under a novel IPTG-inducible promoter at an ectopic locus while removing the genes from their native locus and studying competence development at different pH values and levels of *comC* and *comCDE* expression (new Figure S7). We find that higher levels of *comC* expression are needed for competence development at lower pH. For the *comCDE* mutant we see a similar pattern although we fail to see competence fully developing at a low pH even at full induction. We think this further confirms our hypothesis that pH affects CSP production/detection since for this mutant (as opposed to the wildtype where *comE* expression is not fixed) the ComE-P/ComE ratio can be strongly lowered with decreasing pH if cells fail to produce/detect CSP at the same rate they do at higher pH values. In addition we have constructed a cleaner CSP deficient mutant (*comC*⁻) to study competence development in response to exogenously added CSP at different pH values This cleaner mutant confirms our previous observations with the *comA*⁻ mutant that lead us to suggest that it is mainly CSP production what is affected by pH.

Finally, we expanded the section in the Supplementary Information regarding the model and we now present a simple mathematical argument showing that in terms of the model implementing pH-dependent CSP production or pH-dependent CSP detection is equivalent (Section C of the model description).

The idea of “cell history” is still not well defined and its physiological/ecological significance is not discussed. From a purely technical point of view, it is obvious that in the experimental system used, inoculation of batch cultures with fully induced cells will activate competence earlier than inoculation with uninduced cells.

We use the term ‘cell history’ to refer to our experimental observations because the reason why cells are induced or not before inoculation is that they experienced *different environments* (in the form of different pHs) during the preculture. The memory of such environments is reflected in the status of their competence machineries and as shown by our paper can determine whether they become competent again. In a more natural context such memory can reflect previous exposure to a combination of different environmental factors like antibiotic stress, phosphate concentration and pH.

Such use of the term ‘cell history’ to refer to differences in cell behavior elicited by previous exposure to different environments is standard in the biological literature (Ingolia and Murray *et al.*, 2002; Ozbudak *et al.*, 2004; Wolf *et al.*, 2008; Lambert and Kussell, 2014). In fact, related to the example given by Reviewer 3, in the seminal paper about multistability of the *lac* operon by Ozbudak *et al.* (2004), the term cell history is used to refer to whether cells were previously induced or not by the addition of a lactose analogue during the preculture.

Nevertheless, we acknowledge that we could explain the term better.

We have now explicitly stated what we mean by cell history the first time we refer to this term in the Results section.

“Bistable systems usually exhibit hysteresis. For this reason, we expected that at pH 7.4 where both the ‘ON’ and ‘OFF’ states are attainable, cell history (i.e. environments experienced in the past), would influence competence induction.”

And in the closing sentence of this section

“Thus, our results show that, as a consequence of the positive feedback involved in CSP production, previous exposure to different environmental conditions can determine whether competence is induced or not by modifying the state of the machinery for CSP production and/or sensing.”

Also we have now changed the first line in the Discussion where we refer to cell history. Note that we had already speculated about the possible significance of such memory of past environments for competence development.

“Furthermore, we show that competence development is history-dependent since past environmental conditions can modify the status of the machinery to produce and respond to CSP determining whether competence switches on or not (Figure 6). Hysteresis in the competence response might be particularly important in the natural niche of the pneumococcus, the human nasopharynx. In particular it is consistent with the observation that there is constitutive upregulation of competence in pneumococcal biofilms during nasopharyngeal colonization³⁴. In this context, once competence is triggered for the first time cells would be primed to rapidly initiate another round of competence.”

The authors state that “little is known about why competence is controlled by an autoinducer peptide like CSP” (p 3, l 65). While this statement implies an evolutionary “purpose” that may be difficult to test, there is nevertheless a reasonable hypothesis worth stating here, namely that the concentration of extracellular DNA from lysed, non-competent cells increases with cell density. This seems so obvious that alternative ideas (“timing device”, “diffusion sensing”) should not completely distract from it and relegate its first mention to the Discussion (p 15).

Note that CSP would only allow bacteria to estimate the density of competent cells because non-competent cells do not release CSP. Competent cells in principle would not lyse since they are immune to the killing factors released by other competent cells. Nevertheless the density of competent cells could be a proxy of the density of non-competent cells. We have now included a mention to this hypothesis in the introduction since it was indeed one of the first ones regarding the function of CSP:

“CSP has been classically thought to be a QS signal¹⁰, whose function

could be to monitor the density of potential DNA donors¹⁴.”

Reviewer #3 (Remarks to the Author):

This manuscript consists of mathematical modeling of the signal transduction pathway that leads to late competence gene expression by *Streptococcus pneumoniae* coupled with a series of in vitro experiments that examine predictions of the model in the context of environmental inputs. The model appears carefully constructed and includes the required participants for induction of competence. Although the model omits other known inputs to make the analyses tractable, it is a valid starting point. The authors focus on the effects of pH, cell density and antibiotics on development of competence. These three external variables have already been established to influence competence signaling, although the mechanisms by which they do so are somewhat poorly understood. Two primary components of the presentation include experiments and interpretations that contradict prior studies that a) demonstrated that CSP may function as a timing device to coordinate competence activation and b) that suggest that CSP may first activate competence within a sub-population of cells, which in turn leads to propagation of competence signaling via CSP and activation of com gene expression across the entire population.

There is a considerable amount of data presented. The analyses using the mathematical models and the experiments that the authors present have been carefully designed and performed, the results are clearly presented, the paper is well written, and the interpretations they offer are reasonable - within the confines of their models and experimental design. There are, however, some deficiencies. There is no data presented to try to explain the differences in the present results and prior studies; for example, the use of different strains and growth conditions could account for the disparate behaviors in this and previous studies.

The authors attempt to provide a unifying theory for the role of CSP to help resolve the debate about whether CSP is a true quorum sensing (QS) molecule by positing that CSP should be considered a QS signal, but only in the context of the cellular (micro)environment. Like some of the experimentation and other conclusions, this proposal is not entirely novel and does not substantially advance the field or create new concepts for QS and/or competence. There were only modest attempts at integrating the findings with the biology, physiology and pathogenesis of the organism, or with similar behaviors reported for related organisms. Some of the more novel and interesting findings, such as the effects of pH and potential bistable behavior elicited at pH 7.3, were not thoroughly investigated.

We thank the reviewer for the helpful comments. We now included more data (see below) to compare our results to previous studies. Importantly, this revised Manuscript now provides, to the best of our knowledge, the first direct evidence for QS in pneumococcal competence, directly demonstrating that cell-cell contact is not required for competence initiation as hypothesized by Prudhomme et al., 2016 Plos Genetics. In the new Figure 3, we show that a clean CSP deficient mutant (*comC*⁻) can still initiate competence when it is grown in the presence of wild type pneumococci. Importantly, *comC* mutants can become competent without touching wildtype cells, demonstrating that CSP from the wild type diffuses over the semi-solid surface and is sensed by *comC* mutants.

Specific Comments Line 28. Probably should be modified to say that “competence induction can be simultaneously influenced by cell density...”

We incorporated this suggestion.

Line 34. Here and throughout, the concept of “cell history” is discussed. In this case, it refers to whether cells were cultured in conditions that allowed for prior activation of com gene expression. Subsequently, the kinetics of com activation is stated to depend on the “history” of the cells. It would probably be more clear and accurate to simply state that the authors are comparing behaviors of cells that are already actively transcribing com genes with cells in which com genes have not yet been induced. Not surprisingly, those that are already induced for com expression show more rapid activation of the pathway when transferred to inducing conditions. One possible analogy could be that if cells are grown on an alternative carbohydrate source (e.g. lactose), a dramatic decrease in the lag phase is observed when those cells are transferred to medium containing lactose, as compared to cells that were first cultured on glucose. In contrast, when one thinks of cell “history”, it seems more common that this is referring to cells that have genetically and physiologically differentiated into an distinct state, such as persists that arise in long-term culture. Thus, it seems more appropriate to replace the concept of the cells’ “history” with using “previously induced” or “uninduced” cells in the experimental systems.

This was a point brought up by Reviewer 2 as well. As we replied to this reviewer, we use the term ‘cell history’ to refer to our experimental observations because the reason why cells are induced or not before inoculation is that they experienced *different environments* (in the form of different pHs) during the preculture. The memory of such environments is reflected in the status of their competence machineries and as shown by our paper can determine whether they become competent again. In a more natural context such memory can reflect previous exposure to a combination of different environmental factors like antibiotic stress, phosphate concentration and pH.

Such use of the term ‘cell history’ to refer to differences in cell behavior elicited by previous exposure to different environments is standard in the biological

literature (Ingolia and Murray *et al.*, 2002; Ozbudak *et al.*, 2004; Wolf *et al.*, 2008; Lambert and Kussell, 2014). In fact, related to the example given by Reviewer 3, in the seminal paper about multistability of the lac operon by Ozbudak *et al.* (2004), the term cell history is used to refer to whether cells were previously induced or not by the addition of a lactose analogue during the preculture.

Nevertheless, we acknowledge that we could explain the term better.

We have now explicitly stated what we mean by cell history the first time we refer to this term in the Results section.

“Bistable systems usually exhibit hysteresis. For this reason, we expected that at pH 7.4 where both the ‘ON’ and ‘OFF’ states are attainable, cell history (i.e. environments experienced in the past), would influence competence induction.”

And in the closing sentence of this section

“Thus, our results show that, as a consequence of the positive feedback involved in CSP production, previous exposure to different environmental conditions can determine whether competence is induced or not by modifying the state of the machinery for CSP production and/or sensing.”

Also we have now changed the first line in the Discussion where we refer to cell history. Note that we had already speculated about the possible significance of such memory of past environments for competence development.

“Furthermore, we show that competence development is history-dependent since past environmental conditions can modify the status of the machinery to produce and respond to CSP determining whether competence switches on or not (Figure 6). Hysteresis in the competence response might be particularly important in the natural niche of the pneumococcus, the human nasopharynx. In particular it is consistent with the observation that there is constitutive upregulation of competence in pneumococcal biofilms during nasopharyngeal colonization³⁴. In this context, once competence is triggered for the first time cells would be primed to rapidly initiate another round of competence.”

Line 50. There is considerable introductory material devoted to diffusion, which was not a variable directly tested in this study. Since diffusibility as a factor in intercellular communication has already been experimentally verified, and is intuitive, this section can be streamlined.

We incorporated this suggestion.

Line 59. Please clarify, is it the role of competence or QS that is the subject of debate?

We refer to the claim that competence is an instance of QS. We clarified this sentence.

“We study pneumococcal competence, a system classically used as an example of QS. However, whether competence is actually controlled by QS has been recently debated.”

Line 73 to end of paragraph. These studies, like the present one, use model systems that are dissimilar to what the organisms face in vivo. This is pointed out not to diminish the importance of any of the studies, but to emphasize that a) pneumococci in a biofilm are exposed to a variety of microenvironments and b) these biofilms contain cells in various stages of growth/growth arrest and are therefore quite different from the cells growing exponentially in planktonic culture. All of this work should therefore be clearly framed in the context of “under the conditions tested”, since competence is exquisitely sensitive to microenvironments and the physiologic status of the cells. Carrying forward this concept throughout the paper, as well as including it when there is disagreement between two studies, would strengthen the presentation.

We now included sentences throughout different sections of the paper that frame the scope of our findings. We also better explore the reasons for the disagreement between our data and previous studies by analyzing new strains.

“Here, we study the regulation of pneumococcal competence by cell density and two environmental factors, antibiotic stress and pH. Using both in vitro experiments and mathematical modeling we study the combined action of these factors on competence induction.”

“This causes a delay in competence initiation, which, at least for our strain, would not result from regulation by a cell-density independent timing device^{15,16,19}.”

“Why is competence controlled by CSP? CSP does not act as a timing device in our encapsulated strain since competence develops in a cell-density dependent manner without the necessity of cell-cell contact^{18,19,22}”

“Note however that the extent to which cells synchronize competence development may vary depending on the strain. In particular, our results suggest that unencapsulated strains may synchronize less, which would explain the difference between the findings reported by Prudhomme et al. (2016) and our study. Decreased synchronization may result from cells keeping more CSP to themselves. In fact, in other species like *S. thermophilus* where competence is controlled by ComS, a peptide that unlike CSP rapidly gets imported back into the cell, the rate of

competence development decreases with the inoculation density³⁹ as observed for the unencapsulated pneumococcal strains. Also, note that *S. pneumoniae* grows primarily in biofilms where there is heterogeneity in the physiological status and microenvironment that different cells experience which can certainly influence the degree of synchronization in competence development across a population.”

We fully agree with the claim that in a biofilm competence development is strongly determined by the microenvironment that the cell encounters. Note that in this respect, our data using microcolonies in agar pads is a first step to better understand competence development in a more nature-like spatial setting.

Line 114. The model that is constructed is fairly simple because the system itself is relatively simple, compared to some other organisms (e.g. Bacillus, certain other streptococci). Still, the authors should acknowledge that other inputs (e.g. ComW, factors that shut off competence) are present and explain how they may be accounted for in their model (e.g. ComX stability).

We now have added a sentence acknowledging that other aspects of competence are not included in the model:

“Since the model is primarily concerned with competence initiation we purposely left out genes crucial for other aspects of competence development (e.g. the stabilizing factor ComW and the immunity gene *comM*)¹⁷ and genes involved in competence shut-off such as DprA²⁵.”

Line 152. Either at this point or in the discussion it would be appropriate to clarify that there are differences in the strain(s) and other variables that might account for the differences between results in citation 18 and this study. As written, it comes across as the authors inferring that the prior work is incorrect, without having done the necessary experiments to take such a position.

We have now tested additional strains to further determine why our results are different from the ones observed by Claverys *et al.* (2006) and Prudhomme *et al.* (2016). We found that a possible source of differences is the fact that our strain is encapsulated whereas all the strains used in their studies lack a capsule. We tested different unencapsulated strains that show a dependency of competence initiation on inoculation density more similar to the results described in these two studies. Nevertheless, we still see that competence can spread in an unencapsulated strain without cell-cell contact being necessary.

“Finally, note that we used an encapsulated strain in contrast to the studies of Claverys *et al.* (2006)¹⁵ and Prudhomme *et al.* (2016)¹⁹ which were based

on unencapsulated strains. We explored whether this could explain our different observations regarding the effect of inoculation density on competence development by studying additional strains: An unencapsulated version of our strain D39, the clinical isolate PMEN14 together with its unencapsulated version and *S. mitis* (unencapsulated). Although we still observe that competence develops later for lower inoculums, for the two capsule knockouts and *S. mitis* the slope of the RLU signal decreases with inoculation density in agreement with the results of Claverys *et al.* (2006) and Prudhomme *et al.* (2016) (Figure S4). A decreasing slope of the RLU signal likely indicates decreasing synchronization in competence development with lower inoculation density. Since cells are synchronized by the common extracellular CSP pool, a possible scenario is that the absence of a capsule impedes cells from exporting all the CSP they produce. This would translate into less synchronization especially at low inoculation densities. In the extreme scenario where cells would keep all the CSP to themselves competence regulation would be in fact independent from cell density. Note, however, that for the capsule knockout of our strain we can both detect CSP in the cell-free supernatant and observe competence propagation in the absence of cell-cell contact (Figure S5). “

We use a different luminometer than the one used by the study of Prudhomme *et al.* Note that we also have discarded that the reason why we obtain different results is the lack of sensitivity of our luminometer. We included a sentence about this in the legend of Figure 2.

“Note that our luminometer has enough sensitivity to detect light from competent cells at a density of 1.52×10^{-3} or higher even if they correspond to a subpopulation (Figure S2). Therefore, competence is not observed earlier because it has not developed yet and not because our luminometer is unable to detect it.”

Line 162. This would be one example of the findings not being entirely novel, since this is a roundabout approach that reaffirms there is a dose-dependent response to CSP by pneumococci.

We do not claim at this point that this aspect is the novelty of the current Manuscript. Rather, we show that pneumococcal competence is triggered by a combination of factors that all act through the action of CSP.

Line 172. It is not clear that there is a “common misconception” related to QS. In reality, most researchers would agree that if the environmental conditions could influence the real or effective concentration of the QS molecule, that you would begin to uncouple QS from cell density.

Some of the main criticism to the QS interpretation of competence is the idea that

cell-density dependence implies the existence of a fixed density threshold (see papers cited in the sentence you're referring to). Also, note that our data and model show that even under the *same* environmental conditions if the inoculum size of two populations is different they will develop competence at different densities.

Line 188-204. Manipulating pH is one way to demonstrate this. Use of cells pre-treated with CSP could further support their hypothesis.

We understand that the reviewer is suggesting an experiment where cells are precultured at pH 6.8 and then incubated with CSP before inoculation to show that this would shorten the initiation of competence. We did such an experiment to study the sensitivity of our luminometer by incubating cells from acid preculture with CSP for 10min before inoculation. Note that with a high enough CSP concentration this time is enough to fully induce competence in these cells as is the case in our experiment. This is not surprising because pretreating cells with CSP eliminates the positive feedback loop inherent to competence development and in particular an important part of the waiting time which comes from the expression and maturation of *comC* into CSP. One could lower the incubation time and the CSP concentration to get intermediate timings that are lower than the waiting time observed without pretreating the cells with CSP but still higher than zero. Nevertheless note that a better source of evidence that cells need some time to build up the machinery of competence expression are our new experiments with the IPTG inducible *comC* mutant (Figure S7). Even when inoculated in media where the IPTG concentration fully induces *comC* expression (1000uM), competence still takes some time to appear (around 100min).

Figure 3. It would be helpful to see this data in a histogram-type display. The issue is whether their data really do refute the idea of a sub-population or if there may be some issue with the methodology. If the model were modified a little, would the data appear more similar to that reported by Prudhomme et al. (Ref. 22)?

Agreed and we have now plotted the data in a histogram display that shows the distribution of fluorescence signal in the population across time. The new version of the figure better illustrates that competence development is homogeneous across all cells in the population and that we do not observe a subpopulation of fully competent cells before competence develops in the entire population since the histograms of fluorescence are always unimodal. Also, we have now added an additional microscopy experiment to Figure 3 where we offer more direct evidence against the hypothesis that CSP is transmitted by cell-cell contact. In this experiment we show that a microcolony formed by cells that do not produce CSP (*comC*- mutant) can become competent by being surrounded by microcolonies of the WT that do produce CSP even in the absence of cell-cell contact.

Figure 3. It seems a little surprising that *S. pneumo* could grow at pH 9.0, or

even much above 8.0, yet according to 4a it does. If it cannot grow at pH 9.0, then one explanation for the data would be that, during the long lag period, the cells are producing acid and lowering the pH to a growth-permissive level. If that is the case, then the data should be reported and the conclusions modified accordingly.

We agree that this might be an explanation for the lag time observed at pH 9.0. Since we ignore acidification by growth in the model (which at this pH would be considerable before competence initiation) and *S. pneumoniae* hardly faces such a high pH in its natural niche, we decided to remove the data for this pH from the figure.

Note that we have now included a sentence in the caption of Figure 4 relative to the issue of acidification by cell growth:

“Note that the indicated pH is the initial pH of the medium which does not stay constant due to acidification by growth (Figure SX). Although the pH drops considerably in fully-grown cultures, acidification is still minor at the density where competence develops.”

Line 235. It is a little perplexing why a *comA* mutant was utilized instead of a *comC* deletion, it seems the latter would yield cleaner data. It also seems there should be consideration given to other explanations for why there are different behaviors of the WT and *comA* knockout other than rate of CSP export, even though the results are consistent with the mathematical model outcome.

We imagine that a potential concern of using a *comA* mutant is that bacteria can still produce ComC (without being able to export it) so they could accumulate it inside and eventually leak it. Note however that in the absence of a positive feedback loop, as a consequence of the *comA* deletion, the levels of *comC* expression will be basal. Nevertheless, we have repeated all experiments where we used the *comA* mutant with a *comC* mutant and the new data confirms our previous results with the *comA* mutant (new figures S1 and S7).

Line 247. It appears the authors are trying to say that cells that are exposed to higher pH do not need to shunt as much ATP to maintaining the delta pH component of the pmf as do cells that are exposed to more acidic conditions; thus the rate of CSP export through ComAB can be higher because the former cells have greater ATP pools. This may or may not be true. While the authors make some concessions to other scenarios for altered CSP production below (line 249-53), it would be helpful to be more specific what they mean vis-à-vis impacts on pmf and the outcome of the experiments.

Indeed the referee is correct that this is mainly based on speculation. We have now made it more clear that this is a possible scenario, also fortified by our new

results regarding the new *comC* mutants.

Line 257. It would strengthen the presentation and allay some of the concerns stated above if some experimental verification of this hypothesis were provided. That is, the authors could use MS to directly quantify CSP secreted by cells under different pH conditions, or use the supernatants derived from cells exposed to different pH values to see how effectively they activated a com-reporter strain (a surrogate measure of CSP concentration in the supernatant fluids).

As we answered to Reviewer 2, who brought up the same point, we have now included a more thorough mutant analysis by placing *comC* and *comCDE* under a novel IPTG-inducible promoter and studying competence development at different pH values and levels of *comC* and *comCDE* expression (Figure S7). We find that higher levels of *comC* expression are needed for competence development the lower the pH. For the *comCDE* mutant we see a similar pattern although we fail to see competence fully developing at a low pH even at full induction. We think this further confirms our hypothesis that pH affects CSP production/detection since for this mutant (as opposed to the wildtype where *comE* expression is not fixed) the ComE-P/ComE ratio can be strongly lowered with decreasing pH if cells fail to produce/detect CSP at the same rate they do at higher pH values. In addition we have constructed a cleaner CSP deficient mutant (*comC*⁻) to study competence development in response to exogenously added CSP at different pH values This cleaner mutant confirms our previous observations with the *comA*⁻ mutant that lead us to suggest that is mainly CSP production what is affected by pH.

Finally, we expanded the section in the Supplementary Information regarding the model and we now present a simple mathematical argument showing that in terms of the model implementing pH-dependent CSP production or pH-dependent CSP detection is equivalent (Section C of the model description).

The section related to antibiotic effects on competence should be tempered somewhat since the actual mechanisms by which these antibiotics are inducing competence are not well-defined. It cannot be excluded that factors related to growth arrest or overlap with other stress regulons could be modifying cell behavior independently of the variables being tested.

We thank the referee for pointing this out. As we answered to Reviewer 1, we have now included a paragraph in this section regarding other mechanisms.

“Finally, note that there might be alternative mechanisms from the ones incorporated in the model by which antibiotic stress affects competence. For instance, antibiotics reduce growth and can induce stress responses that lead to global changes on transcription and translation”

The induction of bi-stable behavior at pH 7.3 but not at 7.4 is perhaps the most intriguing and novel part of the story. While it is not unreasonable to suggest that the system is thresholded, allowing for potential stochastic behaviors, there is insufficient direct evidence. Use of other common technical approaches to prove the existence of responsive and non-responsive populations (e.g. microscopy, cell sorting) would strengthen the hypothesis.

We do find that there is bistable behavior both at pH 7.3 and 7.4, since cells from non-acid preculture remain competent if inoculated at a high enough density. The difference between both pH values is that the threshold density at which cells need to be inoculated is higher for pH 7.3. This is consistent with our previous findings since at a lower pH cells would need a higher amount of initial CSP to remain 'ON' which means that more 'ON' cells (from non-acid preculture) need to be inoculated at the start.

We have now clarified this section of the results.

“We then studied whether there is history-dependence by inoculating cells coming from non-acid preculture at pH 7.4. We found that cells coming from a non-acid preculture became competent when inoculated at densities above $OD_{595nm} 2.4 \times 10^{-4}$, which demonstrates that cell history can influence competence development at this pH (Figure 6c).”

Discussion. The three main issues surfacing in the discussion are related to the use of the concept of “history” of the cells, the tentative nature of the proof of bi-stable behavior, and a lack of citation of literature that is directly related to the study, but in which the work was conducted with organisms other than *S. pneumoniae*. There are a number of quality studies done on environmental effects on competence in related organisms that would be of value to include in the discussion. Likewise, it could be argued that any discussion of competence in streptococci should include some mention of the ComRS pathway.

We have now referred to other studies on the effect of the environment in competence of *S. mutans*.

“Importantly, CSP can integrate additional environmental cues like oxygen availability through the CiaRH two component system, which represses *comC* post-transcriptionally and is required for virulence expression and host colonization^{14,42}. In fact, CiaRH is key for the regulation of competence by multiple environmental signals in other streptococci like *S. mutans*⁴³”

We have now included a reference to the ComRS pathway of *S. thermophilus*. In fact, we think it might be useful to understand the differences between the capsulated and unencapsulated strains.

“Decreased synchronization may result from cells keeping more CSP to themselves. In fact, in other species like *S. thermophilus* where competence is controlled by ComS, a peptide that unlike CSP rapidly gets imported back into the cell, the rate of competence development decreases with the inoculation density³⁹ as observed for the unencapsulated pneumococcal strains.”

REVIEWERS' COMMENTS:

Reviewer #2 (Remarks to the Author):

The authors thoroughly addressed my comments and concerns. They have produced a very solid manuscript. I have no further comments.

Reviewer #3 (Remarks to the Author):

The authors have been very responsive to the critiques, particularly from Reviewers 2 and 3 who asked for quite a bit of modification and inclusion of new experiments. The authors made an earnest effort to perform additional experiments, reframe some of the more controversial interpretations, and to better support their core hypothesis. The debate about "timing device" versus QS molecule is not trivial and their experiments and modeling provide good support for the latter. Overall, then, the manuscript is much improved. I have only a few minor suggestions.

Specific Comments

1. The last paragraph in the introduction can be shortened by removing the description of what is subsequently presented in the results and discussion.
2. Line 130. This is the first reasonable opportunity to clarify what the authors mean when they use the term "cell history". They do a good job of explaining this in their rebuttal. I suggest they define "cell history" here so readers know exactly what is intended as they progress further into the manuscript.
3. Line 197. Change to $OD_{595} = 0.1$ or OD_{595} of 0.1.
4. Discussion. There is a small body of recent literature that the authors have not cited where the effects on competence of pH (in a highly controlled environment) and biofilm growth have been examined in similar streptococci; and this work adds support to their interpretations. Such information could be included, perhaps after line 443.